# Ecdysone steroid hormone remote controls intestinal stem cell fate decisions via the *PPARγ*-homolog *Eip75B* in *Drosophila*

Lisa Zipper, Denise Jassmann, Sofie Burgmer, Bastian Görlich, Tobias Reiff*

Institute of Genetics, Heinrich-Heine-University, Düsseldorf, Germany

**Abstract** Developmental studies revealed fundamental principles on how organ size and function is achieved, but less is known about organ adaptation to new physiological demands. In fruit flies, juvenile hormone (JH) induces intestinal stem cell (ISC) driven absorptive epithelial expansion balancing energy uptake with increased energy demands of pregnancy. Here, we show 20-Hydroxy-Ecdysone (20HE)-signaling controlling organ homeostasis with physiological and pathological implications. Upon mating, 20HE titer in ovaries and hemolymph are increased and act on nearby midgut progenitors inducing *Ecdysone-induced-protein-75B (Eip75B)*. Strikingly, the *PPARγ*-homologue *Eip75B* drives ISC daughter cells towards absorptive enterocyte lineage ensuring epithelial growth. To our knowledge, this is the first time a systemic hormone is shown to direct local stem cell fate decisions. Given the protective, but mechanistically unclear role of steroid hormones in female colorectal cancer patients, our findings suggest a tumor-suppressive role for steroidal signaling by promoting postmitotic fate when local signaling is deteriorated.

*For correspondence: reifft@hhu.de

Competing interests: The authors declare that no competing interests exist.

## Introduction

Reproduction is an energetically costly process triggering multiple physiological adaptations of organs such as liver, pancreas and gastrointestinal tract upon pregnancy in various species (*Hammond, 1997*; *Roa and Tena-Sempere, 2014*). As a part of the hormonal response to mating and increased metabolic energy consumption, the female *Drosophila melanogaster* midgut is remodeled in size and physiology by stimulating intestinal stem cell (ISC) driven epithelial expansion to achieve an even energy balance (*Cognigni et al., 2011*; *Klepsatel et al., 2013*; *Reiff et al., 2015*).

Since the discovery of adult intestinal stem cells (*Micchelli and Perrimon, 2006*; *Ohlstein and Spradling, 2006*), local signaling pathways such as Notch (N), Jak/Stat, EGFR, Wnt/wingless, Insulin-receptor, Hippo/Warts and Dpp-signaling were shown to contribute to intestinal homeostasis under physiological and challenged conditions like bacterial infections (*Miguel-Aliaga et al., 2018*)(and references therein). The midgut epithelium is maintained by ISC giving rise to only two types of differentiated cells: enteroendocrine cells (EE) and absorptive enterocytes (EC) (*Figure 1A*). Pluripotent ISC are able to self-renew or divide asymmetrically into either committed EC precursor cells called enteroblasts (EB) or enteroendocrine precursor (EEP) cells. EEP, upon timely activation of *scute* in ISC, divide once more prior to terminal differentiation yielding a pair of EE (*Chen et al., 2018*; *Micchelli and Perrimon, 2006*; *Ohlstein and Spradling, 2006*; *Ohlstein and Spradling, 2007*). Nine out of ten ISC mitosis give rise to EB specified by N-activation in EB daughters (*Micchelli and Perrimon, 2006*; *Ohlstein and Spradling, 2006*; *Ohlstein and Spradling, 2007*). Post-mitotic EB retain a certain degree of plasticity by: (1) delaying their terminal differentiation through mesenchymal-to-epithelial transition (MET) (*Antonello et al., 2015a*), (2) changing their fate to EE upon loss of the

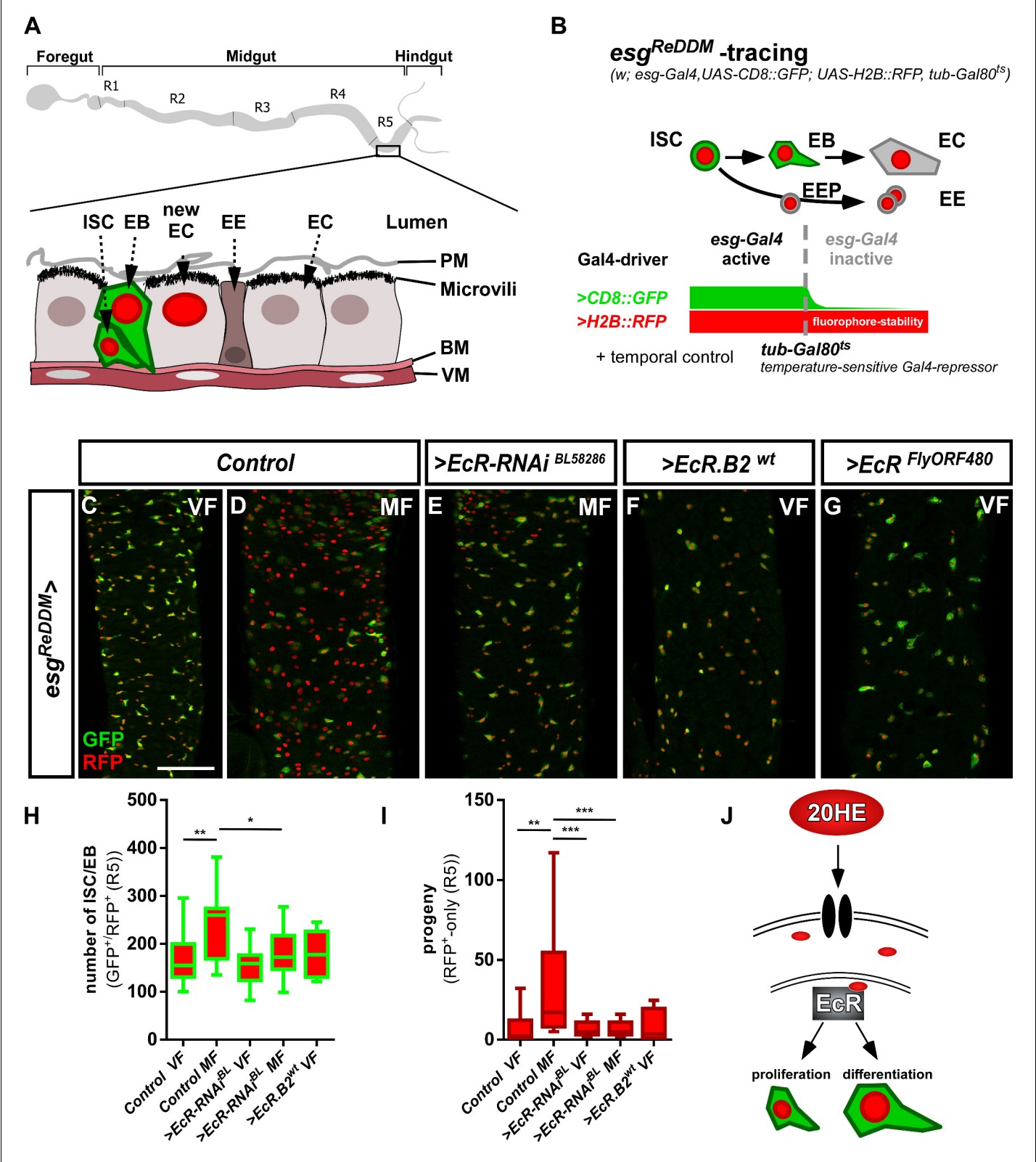

**Figure 1.** The Ecdysone receptor in intestinal progenitors controls tissue homeostasis. (**A**) Scheme of the adult *Drosophila melanogaster* gastrointestinal tract with cartoon depicting the midgut epithelial monolayer composed of intestinal stem cells (ISC), enteroblasts (EB), enterocytes (EC) and enteroendocrine cells (EE) colored according to the lineage tracing system ReDDM with esg-Gal4 (*Antonello et al., 2015a*). (**B**) Schematic of *esg^ReDDM* tracing including full genotype. *ReDDM* differentially marks cells having active or inactive *Gal4* expression. Combined with *esg-Gal4,* active in

*Figure 1 continued*

ISC and EB, *esg^ReDDM* double marks ISC and EB driving the expression of *UAS-CD8::GFP* (membrane CD8::GFP, green), *UAS-H2B::RFP* (nuclear H2B:: RFP, red) and further UAS-driven transgenes (UAS abbreviated as >hereafter in Figure panels). Newly differentiated EC and EE with inactive *esg-Gal4* are RFP$^+$-only owing to protein stability of H2B::RFP. Flies are grown at permissive 18°C in which transgene expression is repressed by ubiquitous tubulin-driven Gal80$^{ts}$. By shifting adult females to the restrictive temperature of 29°C, Gal80$^{ts}$ is destabilized, in turn enabling ReDDM-tracing marking progeny (EE and EC with H2B:RFP nuclear stain) and in parallel manipulation by allowing transactivation of UAS-sequences through *esg-driven* Gal4-expression (*Antonello et al., 2015a*). Posterior midguts (PMG) after seven days of *esg^ReDDM* tracing of control (crossed with *w^1118*) adult MF (D) show mating dependent addition of new EC compared to control VF (C) (*Reiff et al., 2015*). (E) Knockdown of EcR using UAS-driven RNAi abolishes mating induced new EC generation in MF. (F+G) Overexpression of >*EcR.B2* in VF (F) and >*EcRFlyORF^FlyORF840* (G) does not induce proliferation or differentiation of progenitors (ISC+EB). (H–I) Quantification of progenitor numbers (H) and traced progeny encompassing EC and EE (I) in R5 PMG (n = 24,17,17,17, 8). Error bars are Standard Error of the Mean (SEM) and asterisks denote significances from one-way ANOVA with Bonferroni's Multiple Comparison Test (*p<0.05; **p<0.01; ***p<0,001; ****p<0.0001). (J) Cartoon depicting experimental manipulations on EcR signaling pathway investigated with *esg^ReDDM*. Scale bars = 100 μm.

The online version of this article includes the following source data and figure supplement(s) for figure 1:

**Source data 1.** Data from *Figure 1*.
**Figure supplement 1.** The EcR is expressed in the adult *Drosophila* midgut.
**Figure supplement 1—source data 1.** Data from *Figure 1—figure supplement 1*.

transcription factor *klu* (*klumpfuss*) and (3) undergoing apoptosis as an additional homeostatic mechanism (*Korzelius et al., 2019*; *Reiff et al., 2019*).

Apart from aforementioned local signaling pathways, systemic hormones are released into the hemolymph and act on distant organs (*Figueroa-Clarevega and Bilder, 2015*; *Kwon et al., 2015*; *Reiff et al., 2015*). Upon mating, JH released by the neuroendocrine *corpora allata* is able to control ISC proliferation through heterodimers of *Met (methoprene-tolerant)* and *gce (germ cells expressed)* nuclear hormone receptors (*Reiff et al., 2015*). JH coordinates *Drosophila* larval development in concert with the steroid hormone 20-hydroxy-ecdysone (20HE) and both hormones stimulate egg production in adult females (*Bownes et al., 1984*; *Gilbert et al., 2002*; *Kozlova and Thummel, 2000*; *Truman and Riddiford, 2002*). In mated adult female flies, we confirmed an increase of 20HE titers in ovary and detected a similar increase of hemolymph 20HE titers compared to virgin females (*Ameku and Niwa, 2016*; *Gilbert and Warren, 2005*; *Harshman et al., 1999*). The anatomical proximity of ovaries and the posterior midgut (PMG) prompted us to investigate a role for the Ecdysone-receptor (EcR) signaling cascade in organ plasticity during reproduction. Downstream of EcR activation, we detected upregulation of *Ecdysone-induced protein 75B (Eip75B)* protein isoforms ensuring absorptive EC production upon mating. Using the established Notch tumor paradigm, we found that 20HE through Eip75B/PPARγ remote controls EB differentiation and suppresses N-loss of function driven hyperplasia. The mechanism identified in this study not only plays a role in the physiology of mating, but also contributes to our understanding of the protective effects of steroid hormone signaling in the pathophysiology of human colorectal cancer.

## Results

### *EcR* controls intestinal stem cell proliferation and progenitor differentiation

The female fly intestine undergoes various physiological post-mating adaptations including a size increase of the absorptive epithelium (*Cognigni et al., 2011*; *Klepsatel et al., 2013*; *Reiff et al., 2015*). Intrigued by post-mating increases of 20HE titers (*Ameku and Niwa, 2016*; *Harshman et al., 1999*), we explored a role for EcR-signaling in mating adaptations of the adult *Drosophila melanogaster* intestine.

*EcR* encodes for three different splice variants: *EcR.A*, *EcR.B1* and *EcR.B2* (*Cherbas et al., 2003*; *Talbot et al., 1993*), which we detected by PCR in intestinal tissue with highest expression for *EcR. B2* (*Figure 1—figure supplement 1A*). Using EcR antibodies detecting all splice variants, we found EcR in ISC (positive for the Notch ligand Delta$^+$), EB (Notch responsive element, NRE-GFP$^+$) and EC (Discs-large-1, Dlg-1$^+$, *Figure 1—figure supplement 1B–D'''* and *Figure 1A* for an overview). To investigate a role for the EcR in intestinal tissue homeostasis, we first manipulated EcR function in ISC and EB using the 'ReDDM' (Repressible Dual Differential Marker, *Figure 1B*) tracing method to

observe its overall impact on tissue renewal (*Antonello et al., 2015a*). Briefly, *ReDDM* differentially marks cells having active or inactive *Gal4* expression with fluorophores of different stability over a defined period of time. Combined with the enhancer trap *esg-Gal4,* active in progenitors (ISC and EB), *esg^ReDDM* double marks ISC and EB driving the expression of *UAS-CD8::GFP* (*>CD8::GFP*) with short half-life and *>H2B::RFP* with long half-life. Upon epithelial replenishment, CD8::GFP signal is lost and new terminally differentiated EC (Dlg-1$^+$) and EE (Prospero, Pros$^+$) stemming from ISC divisions retain a RFP$^+$-nuclear stain due to fluorophore stability (*Figure 1A,B*; *Antonello et al., 2015a*). Crosses are grown at 18°C in which transgene expression is repressed by ubiquitous tubulin-driven temperature sensitive Gal80$^{ts}$. By shifting adult females to 29°C, Gal80$^{ts}$ is destabilized, in turn enabling spatiotemporal control of *esg^ReDDM*-tracing and additional UAS-driven transgenes in progenitors (*Figure 1B*).

After seven days of *esg^ReDDM*-tracing, mated females (MF, *Figure 1D*) showed increases of progenitor numbers (*Figure 1H*) and newly generated progeny (EE+EC, *Figure 1I*) in the R5 region of the PMG over virgin females (VF, *Figure 1C*) confirming previous observations (*Reiff et al., 2015*). Reducing *EcR*-levels with two different *>EcR* RNAi stocks in MF resulted in a reduction of ISC/EB numbers (*Figure 1E,H*, *Figure 1—figure supplement 1L*) and newly generated progeny (*Figure 1E, I*, *Figure 1—figure supplement 1L*) to levels comparable to VF controls (*Figure 1C,H,I*). We confirmed knockdown efficiency of *>EcR* RNAi in *esg^ReDDM* by measuring fluorescence intensity of EcR in progenitor cells in situ and found EcR-protein levels significantly decreased in both RNAi lines (*Figure 1—figure supplement 1E–H*).

Independent of EcR-abundance, we found that expressing dominant-negative *EcR.B2* isoforms or *EcR*-heterozygosity using *EcR^M554fs*, a well described loss-of-function (LOF) allele, phenocopy *>EcR* RNAi (*Figure 1—figure supplement 1I–K*). Generally, *EcR* LOF leads to a similar phenotype as JH-receptor knockdown in MF (*Reiff et al., 2015*). To investigate whether EcR-levels affect progenitor behavior, we overexpressed wildtype *EcR.B2* and *pan-EcR* using *esg^ReDDM* in VF. We found neither induction of progenitor numbers nor an increase in new EC (*Figure 1F–I*), suggesting that EcR-dependent proliferation and differentiation of progenitors might be limited by 20HE availability (*Figure 1J*).

## Ecdysone titers are increased upon mating and actively transported into progenitors to adapt intestinal physiology

In close anatomical proximity to the PMG, the ovaries are an established ecdysteroidogenic tissue. Determining 20HE titers 48 hr after mating using enzyme immunoassays, we confirmed previous reports of mating dependent increases of ovarian 20HE titers (*Figure 2A*; *Ameku and Niwa, 2016*; *Harshman et al., 1999*) and observed a similar increase in the hemolymph (*Figure 2A*). As a study investigating the *ecdysoneless* mutants suggested that the ovary is the only source for 20HE in adult females (*Garen et al., 1977*), we sought to diminish 20HE titers in adult females. Therefore, we genetically ablated the ovaries using the dominant sterile *ovo^D1* allele in which egg production is blocked prior to vitellogenesis (*Busson et al., 1983*; *Oliver et al., 1987*; *Reiff et al., 2015*; *Figure 2D*). 20HE titers in the hemolymph of *ovo^D1* females are reduced around 40–50% compared to wild-type females (*Figure 2B,A*). Interestingly, 20HE titers in hemolymph and remnants of the ovaries are still significantly increased upon mating of *ovo^D1* females (*Figure 2C*). Consequently, 20HE titers in sterile *esg^ReDDM*/*ovo^D1* MF increase the number of progenitors (*Figure 2E*) and progeny (*Figure 2F*). This suggests that remaining 20HE levels are sufficient to elicit mating related midgut adaptations.

20HE is a polar steroid, which disperses through hemolymph and binds to EcR to activate target gene transcription (*Figure 2D*; *Bownes et al., 1984*; *Gilbert et al., 2002*). Using an established reporter for the activation of 20HE signaling, we found mating induced increases of EcR activity in PMG and ovaries using qPCR (*Figure 2J*). As orally administered 20HE is metabolized and cleared of rapidly, we used the potent non-steroidal EcR agonist RH5849. RH5849 is used as pest control due to its stability, specificity and a 30–60 times higher efficacy compared to 20HE (*Robinson et al., 1987*; *Wing et al., 1988*). Testing concentrations from 1 to 100 µg/ml in timed egg-layings, we observed expected larval molting defects for concentrations from 50 µg/ml upwards (*Figure 2—figure supplement 1A*). Feeding control *esg^ReDDM* VF 50 µg/ml with RH5849, we traced intestinal progenitors with pharmacologically activated EcR-signaling for seven days. RH5849 strongly induces ISC mitosis, reflected by a tenfold increase in newly generated progeny over mating induction

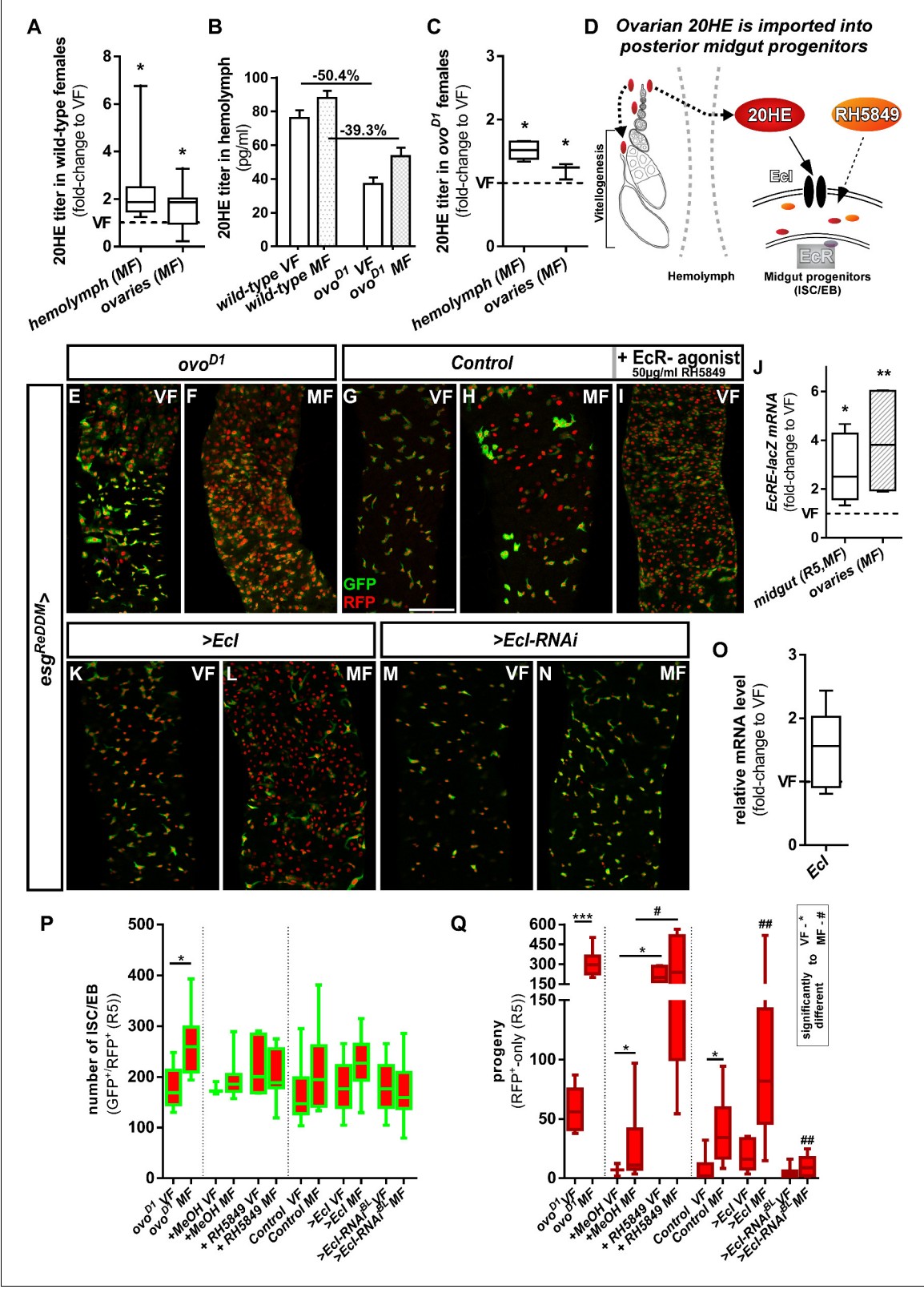

**Figure 2.** Intracellular 20-Hydroxy-ecdysone levels control ecdysone response through the ecdysone importer. (A–C) Determination of 20HE titers in ovaries and hemolymph of wild-type (A,B) and $ovo^{D1}$ (B,C) adult VF and MF 48 hr after mating. (A,C) show fold-change increases over VF titer (dotted line at y = 1). (D) Cartoon depicting ovarian 20HE release to ISC/EB in the adjacent PMG. Please note that in wild-type females, 20HE is incorporated into developing eggs during vitellogenesis, whereas in $ovo^{D1}$ vitellogenesis is absent and might lead to higher (proportional) release into the

*Figure 2 continued on next page*

Figure 2 continued

hemolymph. 20HE from the hemolymph is absorbed by ISC/EB in the PMG, where the cartoon illustrates specific genetical and pharmacological manipulations on the EcR-signaling pathway. (E–I) Representative images of adult PMG after seven days of esg$^{ReDDM}$ tracing of ovo$^{D1}$ VF (E), ovo$^{D1}$ MF (F) and control VF (G) and MF (H). (I) Control VF PMG after oral administration of RH5849 (50 µg/ml). (J) Quantitative RT-PCR on EcRE (Ecdysone responsive elements, [Schwedes et al., 2011]) driving lacZ expression on intestinal cDNA from VF and MF control flies. Values are normalized to VF levels (dotted line at y = 1) and statistically analysed using student´s t-test (*p<0.05, **p<0.01; ***p<0,001;). (K–N) Up- and downregulation of EcI in VF (K,M) and MF (L,N) using UAS driven transgenes after seven days of tracing with esg$^{ReDDM}$. (O) Quantitative RT-PCR of EcI on intestinal cDNA from VF and MF control flies. Values are normalized to VF levels (dotted line at y = 1) and statistically analysed using student´s t-test (*p<0.05, **p<0.01; ***p<0,001;). (P,Q) Quantification of progenitor numbers (P) and traced progeny encompassing EC and EE (Q) in R5 PMG (n = 7,9/3,9,5,13/ 16,13,8,10,10,19). Error bars are Standard Error of the Mean (SEM) and asterisks denote significances from one-way ANOVA with Bonferroni's Multiple Comparison Test (*p<0.05, **p<0.01; ***p<0,001; ****p<0.0001). Scale bars = 100 µm.

The online version of this article includes the following source data and figure supplement(s) for figure 2:

**Source data 1.** Data from *Figure 2*.
**Figure supplement 1.** 20HE regulates physiological adaptations of fatty acid metabolism.
**Figure supplement 1—source data 1.** Data from *Figure 2—figure supplement 1*.

---

(*Figure 2G–I,Q*). Interestingly, RH5849 mediated EcR-activation leads to no accumulation of progenitors (*Figure 2P*) as observed in oncogenic manipulations or disruptions of intestinal homeostasis, suggesting a role for EcR in both, proliferation and differentiation (*Antonello et al., 2015a*; *Chen et al., 2016*; *Patel et al., 2015*; *Reiff et al., 2019*). Given the role of 20HE induced programmed cell death (PCD) in larval metamorphosis and the recently discovered role of EB PCD in adult midgut homeostasis (*Jiang et al., 1997*; *Jiang et al., 2000*; *Reiff et al., 2019*), we addressed PCD by activated caspase-3 staining, but found no increase of PCD by RH5849 (data not shown).

Cellular 20HE uptake was recently shown to depend on *Ecdysone Importer (EcI)* belonging to the evolutionary conserved SLCO superfamily of solute carrier transporters. *EcI* LOF causes phenotypes indistinguishable from 20HE and *EcR* deficiencies in vivo (*Okamoto et al., 2018*). To investigate a role of EcI, we confirmed membrane localization of >EcI tagged with HA by immunostaining in progenitors using esg$^{ReDDM}$ (*Figure 2—figure supplement 1B*). After seven days, forced expression of >EcI using esg$^{ReDDM}$ in VF lead to no increase in progenitor numbers and new EC, underlining that VF 20HE levels are low (*Figure 2K,P,Q*). Further supporting this hypothesis, >EcI in MF lead to an increase of newly generated EC exceeding typical mating induction of MF controls by 4.8 fold (*Figure 2L,P,Q*). Blocking 20HE uptake by *EcI-RNAi* in MF abolished ecdysone induced tissue expansion to VF control levels (*Figure 2N,P,Q*). In addition, we tested a function of the *EcI* gene in mating induction, but found no change in *EcI* expression upon mating (*Figure 2O*). Both, pharmacological and genetic experiments, suggest that 20HE titer and import control EcR-activity upon mating (*Figure 2D*).

Upon mating, absorptive EC undergo metabolic adaptations upregulating genes known for lipid uptake (*Reiff et al., 2015*). We found EC immunoreactive for EcR (*Figure 1—figure supplement 1D*) and investigated a function for EcR in mating related upregulation of lipid uptake by measuring the activity of *sterol regulatory element-binding protein (Srebp)* (*Reiff et al., 2015*). Therefore, we used a GFP-reporter (*Srebp >CD8::GFP*) that is subjected to the same proteolytic processing as Srebp (*Athippozhy et al., 2011*; *Reiff et al., 2015*). Confirming previous observations, *Srebp*-activity increases 2.2 fold upon mating (mean fluorescence intensity, *Figure 2—figure supplement 1C, D*; *Reiff et al., 2015*). Using the *Srebp*-reporter to drive >EcR RNAi in MF, we found reduced *Srebp*-activity comparable VF controls (*Figure 2—figure supplement 1E–F,H*), whereas feeding VF with RH5849 induced *Srebp*-activity 2.4 fold (*Figure 2—figure supplement 1G,H*). Next, we directly addressed lipid uptake with OilRedO-staining on PMG using the EC driver Mex$^{ts}$. In accordance with *Srebp*-activity (*Figure 2—figure supplement 1C,D,G*), feeding flies with RH5849 (*Figure 2—figure supplement 1K*) and forced expression of EcI (*Figure 2—figure supplement 1L*) significantly induced lipid uptake (*Figure 2—figure supplement 1O*) over controls (*Figure 2—figure supplement 1I,J*). Mex$^{ts}$ > EcI RNAi and >EcR RNAi did not result in a reduction of lipid uptake below control levels (*Figure 2—figure supplement 1M,N,I*), confirming previous observations of direct fatty acid incorporation into newly produced eggs (*Reiff et al., 2015*).

These data support the idea that mating related synergistic effects of JH- and 20HE-signaling drive adaptation of intestinal homeostasis and physiology. The JH-signal is transduced by the

transcription factor *Krüppel-homolog 1 (Kr-h1)* in adult MF intestines, which prompted us to explore a function for the 'classical' Ecdysone target genes (*Jindra et al., 2013*; *Reiff et al., 2015*).

## *Ecdysone-induced protein 75b* protein isoforms Eip75B-A and Eip75B-C control enteroblast differentiation

During *Drosophila* development, 20HE pulses lead to direct binding of EcR to regulatory regions of early ecdysone response genes *Ecdysone-induced protein 74A (Eip74EF)* and *Ecdysone-induced protein 75B* (*Eip75B*, *Figure 3A*; *Bernardo et al., 2014*; *Karim and Thummel, 1991*; *Segraves and Hogness, 1990*). First, we performed conventional PCR to analyze JH- and Ecdysone target gene expression. Signal for *Kr-h1-A* (*Kr-h1* in the following), but not *Kr-h1-B*, and for all isoforms of *Eip74EF* and *Eip75B* was found (*Figure 3B*). Using quantitative real time PCR (qPCR) analysis, we confirmed an increase of JH-pathway activity upon mating using *Kr-h1* as control (*Figure 3C*; *Reiff et al., 2015*). To our surprise, we found *Eip74EF* expression unchanged, in contrast to its prominent role in the control of germline stem cell (GSC) proliferation in oogenesis (*Ables and Drummond-Barbosa, 2010*). Instead, we found induction of *Eip75B-A* and *-C* isoforms (*Figure 3C*). *Eip75B* encodes for three protein isoforms (Eip75B-A, -B and –C) that differ in their N-terminal domain structure (*Segraves and Hogness, 1990*) and Eip75B-B lacks one of the two zinc-finger DNA binding domains rendering it incapable of direct DNA binding (*White et al., 1997*).

Intrigued by mating increases of *Eip75B-A /- C* levels, we manipulated Eip75B function using *esg^ReDDM^*. Reducing *Eip75B* levels with RNAi, we found a significant increase in ISC and EB numbers (*Figure 3D,E,L*), but not in EC generation compared to VF controls (*Figure 3M*). In MF, *Eip75B* knockdown reduced newly generated EC compared to MF controls, pointing to a role of *Eip75B* in EB differentiation downstream of EcR-activation (*Figure 3E,M*). Interestingly, feeding flies with RH5849 lacking *Eip75B* in their intestinal progenitors (*Figure 3F*) did not result in newly generated progeny (*Figure 3M* compare to *Figure 2Q*), suggesting a central role for Eip75B in the RH5849 response. Additionally, we investigated homozygous *Eip75B-A* mutants using MARCM clonal analysis with a known LOF allele (*Rabinovich et al., 2016*). MARCM clones of *Eip75B-A* (*Eip75B^A81^*) are significantly larger compared to controls (*Figure 3—figure supplement 1A–D*; *Lee and Luo, 1999*). *Eip75B^A81^* deficient clones contain few differentiated EC (GFP^+^/Dlg-1^+^, *Figure 3—figure supplement 1C*) suggesting disturbed EB to EC differentiation as observed for *esg^ReDDM^* >*Eip75* B-RNAi (*Figure 3E*). Together, these LOF experiments hint to a role for mating induced *Eip75B-A /- C* isoforms in EC differentiation of EB downstream of EcR.

Forced expression of Eip75B variants with *esg^ReDDM^* resulted in three prominent phenotypes: (1) *Eip75B-A* and *Eip75B-C* strongly drive differentiation into EC (Dlg-1^+^, *Figure 3G,J,M*) depleting the entire progenitor pool (*Figure 3G,J,M*). (2) The *Eip75B-B* isoform induces ISC proliferation in VF (*Figure 3I*, *Figure 3—figure supplement 1F*) and raised progenitor numbers suggesting slowed down but not inhibited terminal differentiation (*Figure 3I,L,M*). (3) All Eip75B manipulations strongly reduce the number of newly differentiated EE (*Figure 3—figure supplement 1E*). This suggests that EE differentiation upon *Eip75B-RNAi* and forced expression of the *Eip75B-B* isoform is blocked (*Figure 3I,L*, *Figure 3—figure supplement 1E*). *Eip75B-A* and *Eip75B-C* are sufficient to instantaneously drive progenitors into EC differentiation (*Figure 3G,H,J,K*) even in the absence of 20HE import (*Figure 3H,K*). The immediate strong differentiation stimulus prevents further ISC division and as a consequence also reduces new EE (*Figure 3L,M*, *Figure 3—figure supplement 1E*). Disrupting intestinal homeostasis alters midgut length (*Hudry et al., 2016*) and in line with this, Eip75B manipulations result in a shorter midgut as EC make up around 90% of the whole midgut epithelium (*Figure 3—figure supplement 1G*). Eip75B affects intestinal homeostasis by either slowing down terminal EB to EC differentiation (*Eip75B-B* and *Eip75B-RNAi*) or depleting the ISC and EB pool (*Eip75B-A /– C*, *Figure 3—figure supplement 1G*) preventing sufficient EC production along the gut. When EC production is blocked using >*N* RNAi, the midgut shrinks around one third over a period of seven days (*Chen et al., 2018*; *Guo and Ohlstein, 2015*; *Micchelli and Perrimon, 2006*; *Ohlstein and Spradling, 2006*; *Ohlstein and Spradling, 2007*; *Patel et al., 2015*). Thus, all Eip75B manipulations ultimately lead to an insufficient number of new absorptive EC resulting in a shorter midgut.

*Eip75B* is a long term predicted *PPARγ*-homologue (*peroxisome proliferator-activated receptor gamma*) in *Drosophila*. A close functional homology got recently confirmed in pharmacogenetic approaches demonstrating that *Eip75B*-mutant flies are irresponsive to PPARγ activating drugs

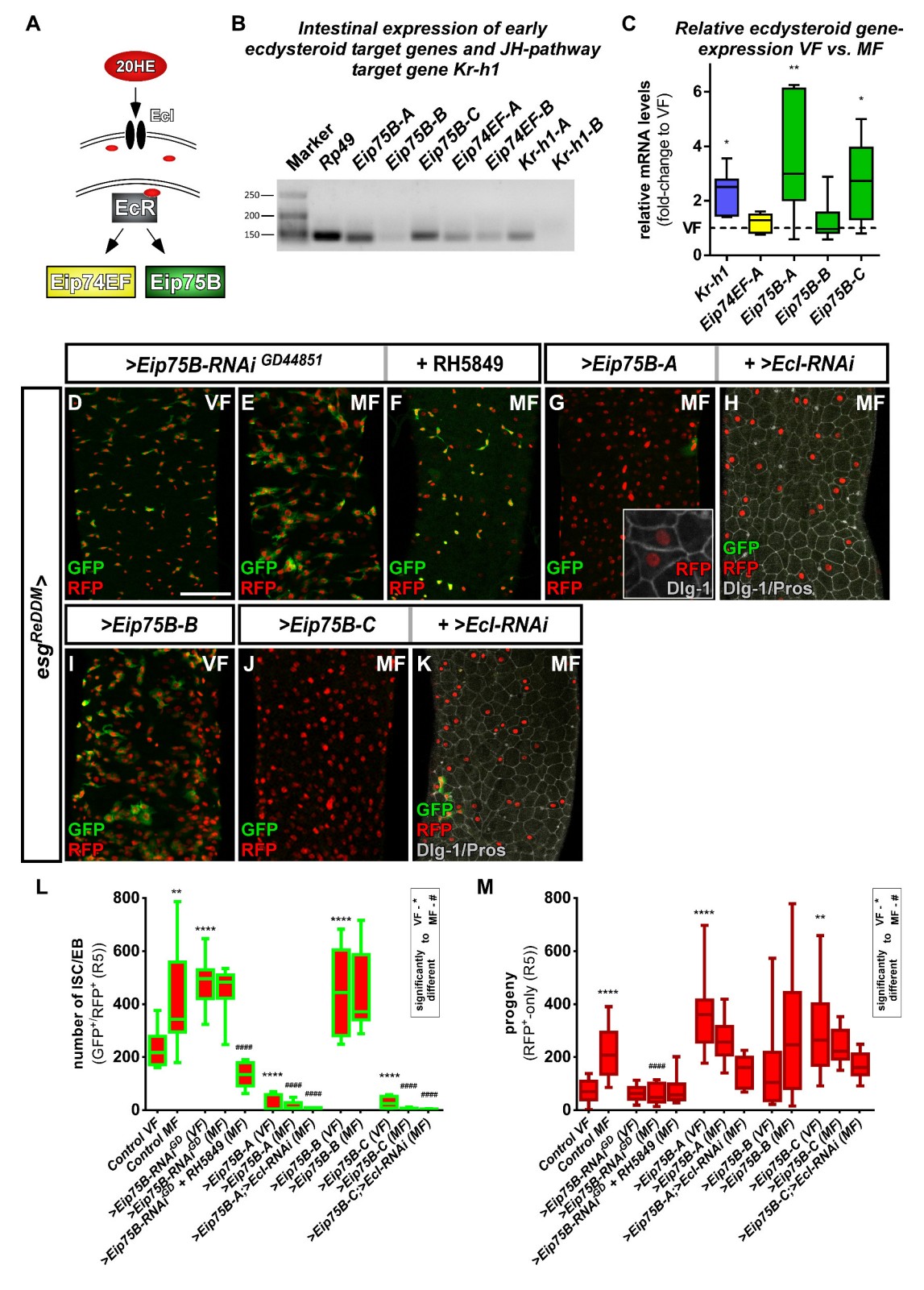

**Figure 3.** *Ecdysone induced protein 75B* is upregulated upon mating and controls progenitor differentiation. (**A**) Cartoon depicting EcR signaling cascade activating early ecdysteroid target genes. (**B**) Expression analysis of Ecdysone- and JH-signaling target genes including protein isoforms on cDNA transcribed from mRNA isolations from whole midgut dissections of MF. (**C**) Quantitative RT-PCR on early ecdysteroid genes on intestinal cDNA from VF and MF control flies. Values are normalized to VF levels (horizontal line = 1) and statistically analyzed using student´s t-test (n = 6; *p<0.05,
*Figure 3 continued on next page*

Figure 3 continued

\*\*p<0.01;). (D–E) RNAi-mediated downregulation of *Eip75B* in VF (D), MF (E) and MF fed with RH5849 (F) after seven days of tracing with *esg*[ReDDM]. (G–K) Representative images of adult PMG with forced expression of *Eip75B* isoforms *Eip75B-A* (G), *Eip75B-A* and *Ecl-RNAi* (H), *Eip75B-B* (I), *Eip75B-C* (J), *Eip75B-C* and *Ecl-RNAi* (K) after seven days of tracing with *esg*[ReDDM]. Inset in (F) depicts epithelial integration of newly generated Dlg-1[+]/RFP[+]-EC. (I–J) Quantification of progenitor numbers (I) and traced progeny encompassing EC and EE (J) in R5 PMG (n = 12,13,10,12,11,11,14,8,10,10,10,5,11). Error bars are Standard Error of the Mean (SEM) and asterisks denote significances from one-way ANOVA with Bonferroni's Multiple Comparison Test (\*p<0.05, \*\*p<0.01; \*\*\*p<0,001; \*\*\*\*p<0.0001, identical p-values are marked by # when compared to MF). Scale bars = 100 µm.

The online version of this article includes the following source data and figure supplement(s) for figure 3:

**Source data 1.** Data from *Figure 3*.
**Figure supplement 1.** Analysis of *Eip75B-A* MARCM clones.
**Figure supplement 1—source data 1.** Data from *Figure 3—figure supplement 1*.
**Figure supplement 2.** EB specific genetic manipulation of Eip75B using *klu*[ReDDM].
**Figure supplement 2—source data 1.** Data from *Figure 3—figure supplement 2*.

---

(*Joardar et al., 2015*; *King-Jones and Thummel, 2005*). This homology is of particular interest, as human PPARγ plays a role in i) pregnancy related adaptations of lipid metabolism (*Waite et al., 2000*) and ii) as target in colorectal cancer (CRC) (*Sarraf et al., 1999*). Thus, we tested pharmacological activation of Eip75B/PPARγ with the established agonist Pioglitazone, a drug used in Diabetes mellitus treatment (*Gillies and Dunn, 2000*; *Jafari et al., 2007*). Flies fed with food containing Pioglitazone show a strong increase in the number of progenitor cells (*Figure 4B,E*) and newly generated EC (*Figure 4F*) compared to controls (*Figure 4A*). Strikingly, knockdown of Eip75B led to irresponsiveness to Pioglitazone (*Figure 4C,D,F*) suggesting that Pioglitazone activates Eip75B in intestinal progenitors.

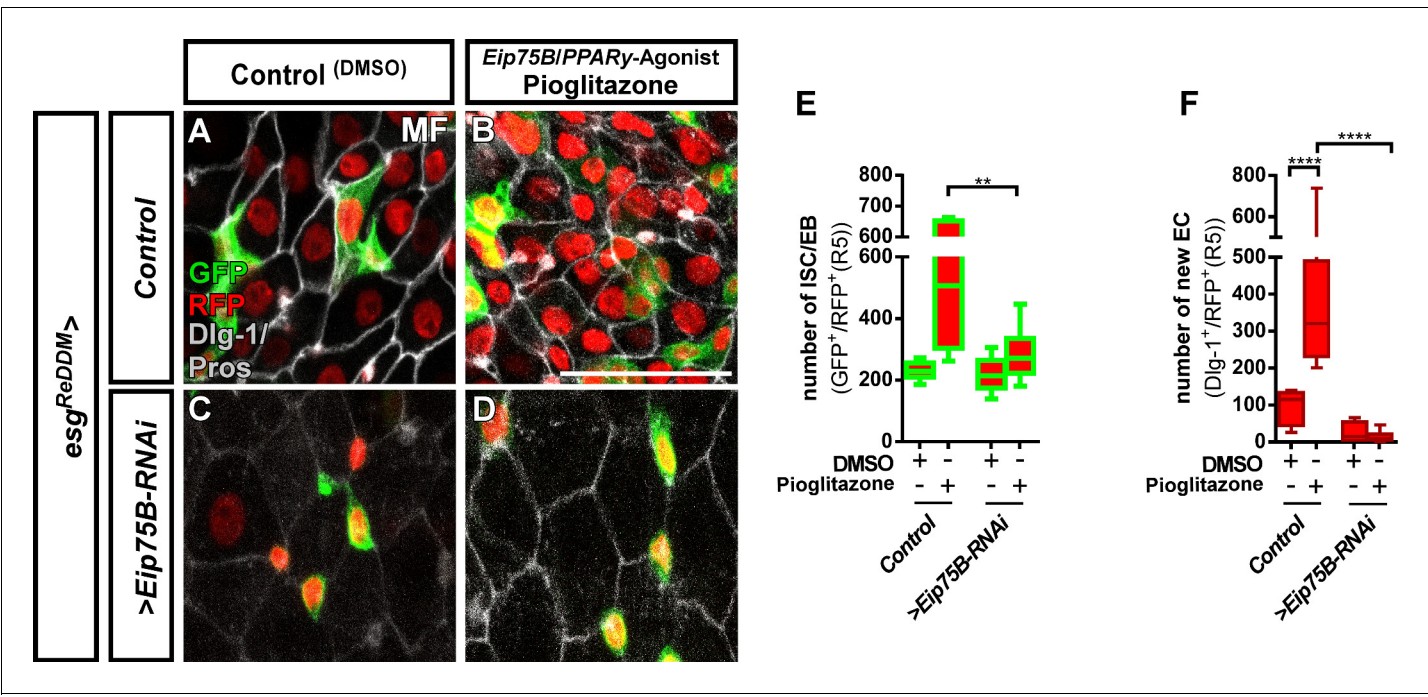

**Figure 4.** The Eip75B/PPARγ agonist Pioglitazone acts through Eip75B to stimulate progenitor differentiation. (A–D) Representative images of adult PMG after seven days of *esg*[ReDDM] tracing of control MF (A,B) and >*Eip75* B-RNAi MF (C,D) fed with DMSO as control (A,C, 2.5 µl/ml food) and Pioglitazone (B,D; 0,002 mg in DMSO/ml food). (E+F) Quantification of progenitor numbers (E) and newly generated EC (F) in R5 PMG (n = 7,6,10,11). Error bars are Standard Error of the Mean (SEM) and asterisks denote significances from one-way ANOVA with Bonferroni's Multiple Comparison Test (\*p<0.05, \*\*p<0.01; \*\*\*p<0,001; \*\*\*\*p<0.0001,. Scale bars = 100 µm.

The online version of this article includes the following source data for figure 4:

**Source data 1.** Data from *Figure 4*.

Taken together, these findings strongly suggested a role for *Eip75B* in EB differentiation. EB are specified by N-signaling and their lineage is maintained by the transcription factor *klumpfuss* (*klu*). *klu*⁺-EB retain some plasticity to change fate to EE upon reduction of *klu*-levels (***Korzelius et al., 2019***; ***Reiff et al., 2019***). We used *klu*^ReDDM^ to explore the function of *EcR* and *Eip75B* in EB lineage identity (***Figure 3—figure supplement 2A***). No changes in EB fate decisions towards EE differentiation were observed upon knockdown of *EcR* and *Eip75B* (***Figure 3—figure supplement 2A,B–D***). *Eip75B-A* and *Eip75B-C* expression with *klu*^ReDDM^ phenocopied EB to EC differentiation effects observed in *esg*^ReDDM^ (***Figure 3—figure supplement 2G,I,J,K***) without non-autonomously inducing proliferation in wild-type ISC of VF and MF (***Figure 3—figure supplement 2J,K***). Expressing *Eip75B-B* in *klu*^ReDDM^ lead to delayed EB differentiation similar to *esg*^ReDDM^ and non-autonomous induction of ISC mitosis (***Figure 3I***, ***Figure 3—figure supplement 2J,L***; ***Patel et al., 2015***; ***Reiff et al., 2019***). Having identified *Eip75B-A /- C* as mating induced differentiation effector of 20HE signaling, we hypothesized and investigated a synergism of 20HE and JH hormonal signaling pathways controlling epithelial expansion upon mating.

## The interplay between JH and 20HE in intestinal progenitors

Our data suggest that mating induced JH and 20HE signaling affect ISC proliferation and EB differentiation through their effectors *Kr-h1* and *Eip75B-A /- C* (***Figure 5K***; ***Reiff et al., 2015***). In VF lacking mating induction of both hormones (***Figure 2A–C***; ***Ameku and Niwa, 2016***; ***Harshman et al., 1999***; ***Reiff et al., 2015***), forced >*Kr* h1 expression doubles progenitor numbers reproducing previous results (***Figure 5B,I***; ***Reiff et al., 2015***). Initially, we aimed to perform a full genetic epistasis analysis for *Eip75B* and *Kr-h1*, but this analysis was hampered as we failed to recombine *Kr-h1-RNAi* with *Eip75B* isoform expressing stocks.

However, we were able to analyze midguts of MF with forced *Kr-h1*-expression and simultaneous *Eip75B-RNAi* (***Figure 5C,D***), which increases progenitor numbers (***Figure 5I***) with only few newly generated EC (***Figure 5J***). This finding supports the requirement of *Eip75B* in EB differentiation (***Figure 3E***) and *Kr-h1* in ISC proliferation (***Figure 5A***). Simultaneous double knockdown of >*Kr-h1-RNAi/Eip75B-RNAi* (***Figure 5H***) in MF did not show mating induction. Progenitor numbers and differentiated progeny are reduced (***Figure 5I,J***) phenocopying single >*Kr* h1-RNAi (***Reiff et al., 2015***). Surprisingly, we observed a strong increase of progenitor numbers forcing *Kr-h1* and *Eip75B-B* expression (***Figure 5F,I,J***). These results indicate an additive role of forced *Eip75B-B* and *Kr-h1*-expression on ISC proliferation, whereas the latter transduces the mating related proliferation response (***Figure 3C***). The stimulus inducing *Eip75B-B* expression yet needs to be identified (***Figure 5K***).

Most interestingly, *Eip75B-A* and –*C*, despite of raised *Kr-h1*-levels, drive progenitors into EC differentiation largely phenocopying sole *Eip75B-A* and -*C* expression (compare ***Figures 3G,J*** and ***5E, G,I,J***). Our data on *Eip75B-A* and -*C* corroborate a potent role for Ecdysone-induced EB differentiation. ISC fate decisions in the adult midgut rely on N-signaling (***Micchelli and Perrimon, 2006***; ***Ohlstein and Spradling, 2006***; ***Ohlstein and Spradling, 2007***). During *Drosophila* follicle cell development N opposes EcR-function (***Sun et al., 2008***), which prompted us to investigate *Eip75B-A /- C* as effectors of EcR-signaling in the context of N-dependent EB differentiation.

## Ecdysone signaling through *Eip75B-A* and *Eip75B-C* controls enteroblast differentiation in a notch deficiency tumor model

In the *Drosophila* and mammalian intestinal stem cell niche, N-signaling specifies EE and EC fate (***Jensen et al., 2000***; ***Micchelli and Perrimon, 2006***; ***Ohlstein and Spradling, 2006***; ***Ohlstein and Spradling, 2007***; ***VanDussen et al., 2012***). In *Drosophila*, mitosis of *N* mutant ISC generates only ISC-like progenitor cells and EE, which results in an intestinal epithelium accumulating a lack of newly produced EC and compensatory proliferation (***Figure 6A,B***; ***Chen et al., 2018***; ***Guo and Ohlstein, 2015***; ***Micchelli and Perrimon, 2006***; ***Ohlstein and Spradling, 2006***; ***Ohlstein and Spradling, 2007***; ***Patel et al., 2015***).

Using N-LOF tumors as an experimental paradigm lacking EC production, we sought to investigate the differentiation inducing properties of Eip75B-A /- C. Reduction of N by RNAi or a dominant-negative N-receptor using *esg*^ReDDM^ lead to tumors of different sizes, which we quantified and classified according to their number (***Figure 6B***). We predominantly found tumors of 5–10 cells

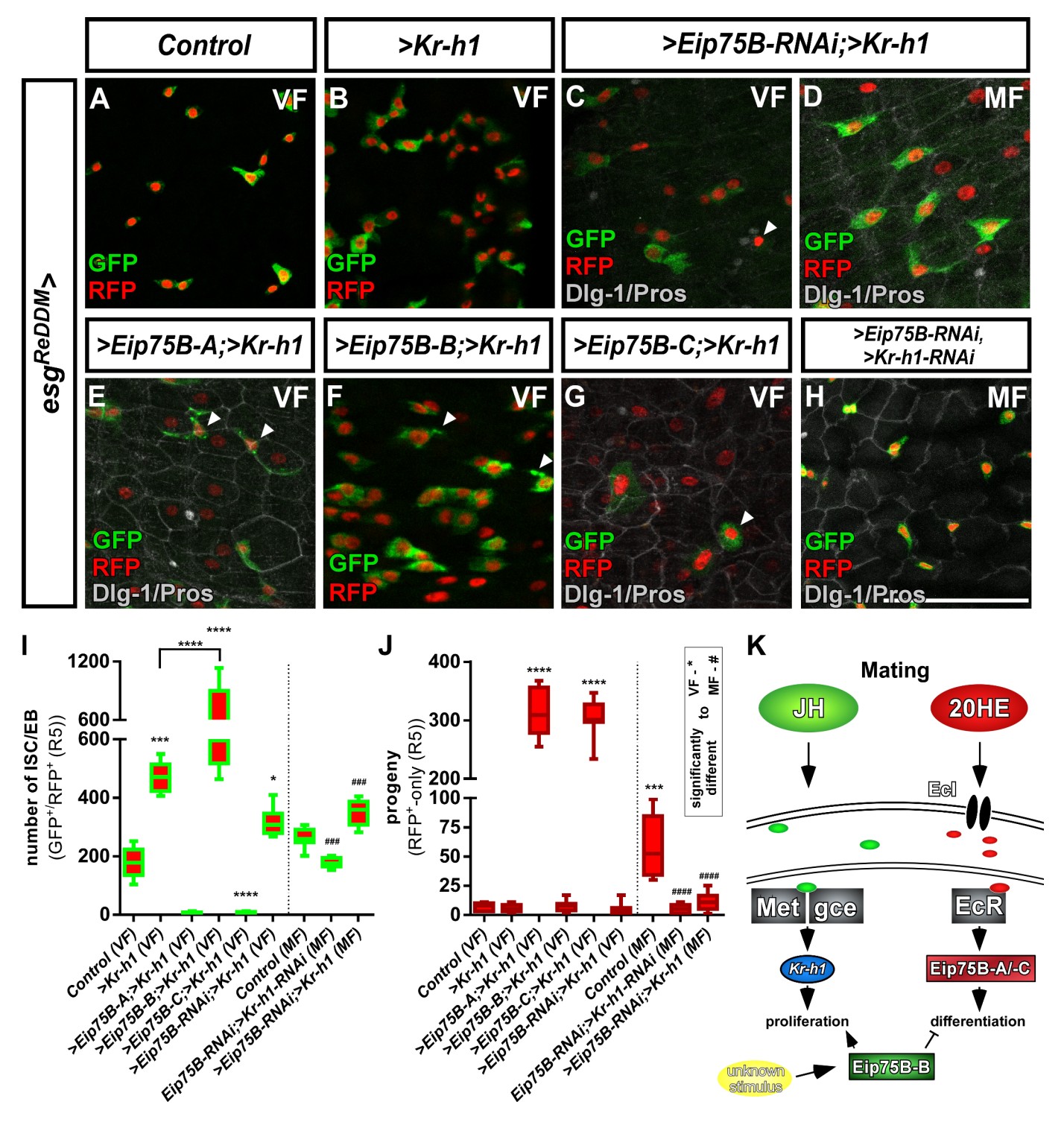

**Figure 5.** Crosstalk between JH- and Ecdysone-signaling pathways controlling intestinal progenitor proliferation and differentiation. (A–B) Images of adult PMG of control VF (A) and forced expression of > *Kr* h1 (B) traced for seven days with *esg*^ReDDM^. (C–H) Images of adult PMG with forced expression of > *Kr* h1 with > *Eip75* B-RNAi in VF and MF (C,D) and expression of *Eip75B* isoforms > *Eip75* B-A (E),>*Eip75* B-B (F),>*Eip75* B-C (G) and double > *Eip75B-RNAi*/>*Kr-h1-RNAi* (H) after seven days of tracing with *esg*^ReDDM^. Please note that genotypes of (C–G) were accompanied with semi-lethality even at permissive 18°C, suggesting for example background transgene expression or position effects most probably caused by the total number of six transgenes including *esg*^ReDDM^. PMG of > *Kr* h1 combinations also showed some progenitor lethality indicated by membrane-blebbing and irregularities (arrowheads, (E–G) as described in **Reiff et al., 2019**. (J) Quantification of progenitor numbers (I) and traced progeny encompassing

*Figure 5 continued on next page*

Figure 5 continued

EC and EE (J) in R5 PMG (n = 12,5,9,8,7,13,8,8,13). Error bars are Standard Error of the Mean (SEM) and asterisks denote significances from one-way ANOVA with Bonferroni's Multiple Comparison Test (*p<0.05, **p<0.01; ***p<0,001; ****p<0.0001, identical p-values are marked by # when compared to MF). Scale bars = 50 μm (K) Cartoon depicting transcriptional effectors of JH- and Ecdysone signaling pathways. The JH receptor is formed by a heterodimer of *Methoprene tolerant (Met)* and *germ cells expressed (gce)*. Ligand bound receptor activates the transcription of *krüppel homolog 1 (Kr-h1)* mediating mating effects in the adult intestine (*Reiff et al., 2015*) Forced expression of Eip75B-B affects proliferation and differentiation through an yet unknown stimulus.

The online version of this article includes the following source data for figure 5:

**Source data 1.** Data from *Figure 5*.

(*Figure 6B*, *Figure 6—figure supplement 1A*), but also few tumors containing more than 100 ISC/ EE-like cells with indistinguishable single or multiple ISC origin (*class IV*, *Figure 6B*, *Figure 6—figure supplement 1A*).

Co-expression of *Eip75B-A /- C* with N-LOF strongly reduced total tumor number (*Figure 6C,D, F*) and smaller *class I, II* and *III* tumors (*Figure 6—figure supplement 1A*), whereas *Eip75B-RNAi* results in confluent *class IV* tumors all along the PMG (*Figure 6E*). Although tumor number is strongly reduced (*Figure 6F*), occasional *class IV* tumors (<1/PMG) are able to escape suppressive Eip75B-A /- C showing no reduction in tumor size (*Figure 6—figure supplement 1A*). Additional tumorigenic mechanisms such as Upd- or EGF-ligand upregulation may counteract the tumor suppressive function of 20HE signaling in *class IV* tumors (*Patel et al., 2015*).

Since Eip75B-A and Eip75B-C re-enable EC differentiation despite the lack of N (*Figure 6C,D*; Dlg-1$^+$/RFP$^+$), we investigated whether the EcR-pathway and pharmacological Eip75B-activation are able to trigger EC fate in a N-LOF context. Activation of Eip75B using Pioglitazone led to more newly generated EC (*Figure 6H,I*) compared to controls (*Figure 6G*). In addition, we observed a slight, but significant reduction in the number of N-tumors upon feeding Pioglitazone (*Figure 6J*).

Given the protective role of steroid hormone signaling in colorectal cancer (CRC), we sought to investigate N tumor numbers upon EcR-signaling activation (*Chen et al., 1998*; *Hendifar et al., 2009*; *Lin et al., 2012*). We activated EcR-signaling with RH5849 (*Figure 7B*) or by raising intracellular levels of 20HE expressing > EcI in esg$^{ReDDM}$(*Figure 7D*). Either manipulation resulted in numerous confluent tumors reflecting the previously observed role of 20HE in proliferation (*Figure 7F*). In line with this,>*EcI* RNAi- and >*EcR* RNAi significantly reduced tumor burden (*Figure 7C,E,F*).

More importantly, ecdysone-pathway activation led to significantly higher numbers of newly generated EC (arrowheads in *Figure 7B,D,G*), suggesting an upregulation of *Eip75B-A* and *Eip75B-C* in Notch tumors upon EcR-activation through RH5849 and > *EcI*. To exclude remaining N activity, we generated MARCM clones mutant for *N* (N$^{55e11}$) (*Guo and Ohlstein, 2015*). In agreement with our previous observations, activating 20HE signaling in N clones led to a tenfold increase in EC numbers over controls (*Figure 7—figure supplement 1A–C*) and forced expression of > E75 A/-C to instant EC formation (*Figure 7—figure supplement 1D–E*). Interestingly, in the presence of N, its activity is increased upon RH5849 feeding (*Figure 7—figure supplement 1F*), which suggests an interplay between Notch and EcR-signaling that might converge on *Eip75B* (this study) and *klu* (*Korzelius et al., 2019*; *Reiff et al., 2019*).

Taken together, our data establish 20HE as a systemic signal controlling reproductive adaptations in synergism with JH (*Figure 7H*). Focusing on EcR-signaling in ISC and EB, we found the mating induced *Eip75B-A /- C* isoforms acting as differentiation factors in EB. Eip75B-A /- C induces EC fate even in Notch-deficient midgut progenitors, an experimental paradigm used to recapitulate early steps of tumorigenesis. Notch dependent fate decisions in midgut precursor cells are conserved between flies and humans. Thus, our findings suggest a new mechanism how steroid hormone signaling might suppress tumor growth by promoting post-mitotic cell fate in intestinal neoplasms marked by a lack of fate specification.

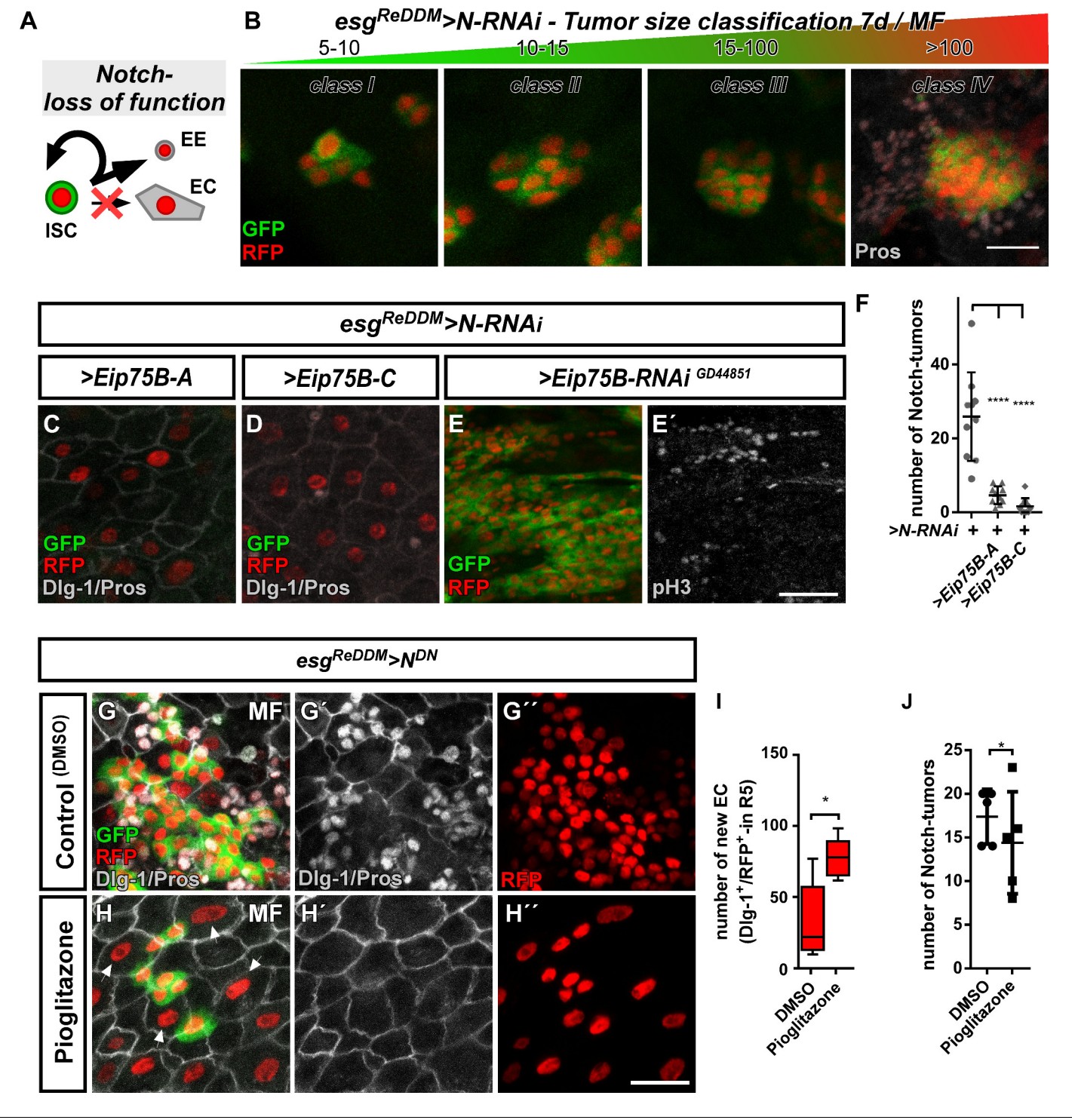

**Figure 6.** *Ecdysone induced protein 75B* promotes EB differentiation in a Notch tumor paradigm. (**A**) Cartoon depicting cell fate changes upon N-LOF combined with *esg*[ReDDM] coloring (***Ohlstein and Spradling, 2006***; ***Ohlstein and Spradling, 2007***). (**B**) > *N* RNAi driven by *esg*[ReDDM] leads to different tumor sizes after seven days of tracing that were classified in four classes according to size. Note, ISC-like/EE clusters up to four cells are not quantified as they occasionally occur in controls too. Progenitors are double labelled (GFP[+]/RFP[+]), where newly generated EE are identified by immunostaining for Prospero (Pros) and H2B::RFP trace from *esg*[ReDDM]. (**C–D**) Co- expression of *Eip75B* isoforms > *Eip75* B-A (**C**) and > *Eip75* B-C (**D**) after seven days of tracing with *esg*[ReDDM] > *N* RNAi in MF. Note additional Dlg-1[+]-immunoreactivity in (grey,C+D) demonstrating epithelial integration of newly generated cells as EC (Dlg-1[+]/RFP[+]). (**E+E'**) RNAi-mediated downregulation of *Eip75B* in MF shows confluent N-tumors (**E**) accompanied by increased mitosis (**E'**). (**F**) Quantification of ISC progeny encompassing ISC-like and EE total N-tumor number in R5 PMG (n = 10,10,10). (**G–J**) Feeding of the
*Figure 6 continued on next page*

*Figure 6 continued*

Eip75B/PPARγ agonist Pioglitazone to *esg^ReDDM^ > N^DN^* flies leads to the generation of newly generated EC ((**H,I**), Dlg-1$^+$/RFP$^+$) after seven days of tracing compared to DMSO controls (**G,I**). (**I–J**) Quantification of ISC progeny encompassing ISC-like and EE total N-tumor number in R5 PMG (n = 5,5). Error bars are Standard Error of the Mean (SEM) and asterisks denote significances from one-way ANOVA with Bonferroni's Multiple Comparison Test (*p<0.05, **p<0.01; *** p$$BOX_TXT_END$$. <0,001; ****p<0.0001). Scale bars = 25 μm.

The online version of this article includes the following source data and figure supplement(s) for figure 6:

**Source data 1.** Data from *Figure 6*.
**Figure supplement 1.** Size distribution of different Notch-tumor classes.
**Figure supplement 1—source data 1.** Data from *Figure 6—figure supplement 1*.

## Discussion

### Systemic hormones and intestinal remodeling upon mating

Growing offspring involves significant metabolic adaptations to raised energy demands in mothers. *Drosophila melanogaster* uses a classical r-selected reproductive strategy with high number of offspring and minimal parental care (*Pianka, 1970*). Upon mating, egg production is increased tenfold supported by metabolic and behavioral adaptations such as food intake and transit, nutrient preference and a net increase of absorptive tissue in the PMG depending on JH release (*Carvalho et al., 2006*; *Cognigni et al., 2011*; *Reiff et al., 2015*; *Ribeiro and Dickson, 2010*).

Here, we describe a role for the steroid-hormone ecdysone controlling intestinal tissue remodeling upon mating through EcR-signaling. The unliganded heterodimer of EcR and ultraspiracle (USP) binds its DNA consensus sequences serving as a repressor. Upon ligand binding, the EcR/USP heterodimer turns into an activator inducing the expression of early response genes *Eip74EF*, *Eip75B* and *broad (br)* (*Schwedes et al., 2011*; *Uyehara and McKay, 2019*). Initially, we investigated knockdown of *USP, br* and *Eip74EF* in intestinal progenitors and observed effects on progenitor survival independent of mating status (*Figure 3C*, Zipper and Reiff, *unpublished data*).

Our data support the idea that 20HE and JH synergistically concert intestinal adaptations balancing nutrient uptake to the increased energy demands after mating (*Figures 1–5*). We confirmed that mating induces ovarian ecdysteroid biosynthesis stimulating egg production by ovarian GSC (*Figure 2A*; *Ables and Drummond-Barbosa, 2010*; *Ameku and Niwa, 2016*; *Ameku et al., 2017*; *Harshman et al., 1999*; *König et al., 2011*; *Morris and Spradling, 2012*; *Uyehara and McKay, 2019*). In accordance with an inter-organ signaling role for 20HE, we also detected increased 20HE titers in the hemolymph (*Figure 2A*). Surprisingly, we found this mating dependent increase of 20HE titers still present upon partial genetic ablation of the ovaries using *ovo^D1^* (*Figure 2B*; *Reiff et al., 2015*). This can be explained by either (i) other source(s) for 20HE in adult females like the brain (*Chen et al., 2014*; *Itoh et al., 2011*) or (ii) 20HE release from remaining germarial cells in stage 1–4 eggs of *ovo^D1^* VF and MF. Indeed, germarial cells have been shown to express the ecdysteroidogenic *Halloween* genes (*Ameku and Niwa, 2016*; *Ameku et al., 2017*). In addition, *ovo^D1^* MF lack ovarian 20HE uptake during vitellogenesis (*Figure 2D*) of later egg stages, which potentially contributes to 20HE titer in the hemolymph of *ovo^D1^* females and intestinal epithelium expansion (*Figure 2D,E,J,K*; *Enya et al., 2014*).

The epithelial expansion upon mating of *ovo^D1^* females has been observed before (*Reiff et al., 2015*) and might be comparably high due to the different genetic background of *ovo^D1^* flies compared to *w^1118^* (*Figure 2Q*). While this work was in review, genetic ablation of ovarian ecdysteroidogenic enzymes was performed in female flies and resulted in a reduction of mitotic ISC. However, a direct assessment of 20HE titers was not performed and a mating dependent context was not analyzed (*Ahmed et al., 2020*). Taken together, current data cannot clearly dissect whether the ovary is the exclusive source of 20HE (*Garen et al., 1977*). In future experiments, 20HE titers have to be directly addressed to investigate whether other sources like the brain might be involved in 20HE release upon mating and other physiological stimuli (*Chen et al., 2014*; *Itoh et al., 2011*).

In the adjacent posterior midgut, we describe the impact of 20HE on *Eip75B-A /- C* expression (*Figure 7H*) and show that 20HE import triggers EcR-activation: i) in VF with low ovarian 20HE levels, overexpression of wild-type *EcR* and *EcI* are unable to elicit neither ISC proliferation nor differentiation effects (*Figures 1–2*). ii) pharmacological activation through RH5849 and Pioglitazone (*Figures 2*

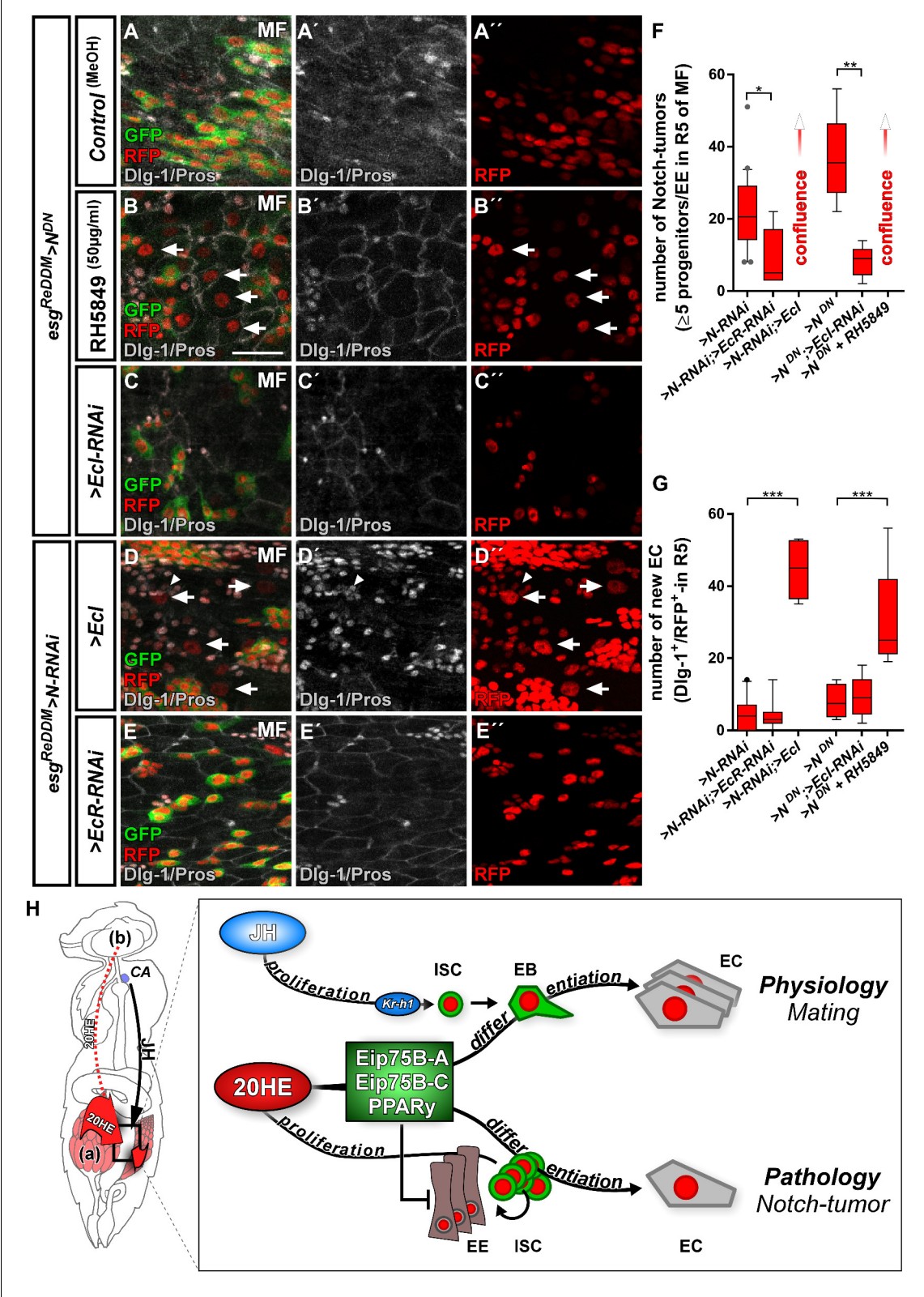

**Figure 7.** Ecdysone signaling promotes EC-fate in a Notch tumor paradigm. (**A–E′′**) Pharmacological and genetic manipulation of Ecdysone signaling. Adult PMG of MF with N-LOF (>N RNAi or > $N^{DN}$) treated with MeOH as control (**A–A′′**) or RH5849 (**B–B′′**) to activate Ecdysone signaling. Arrowheads highlight newly generated Dlg-1+/RFP+-EC (B+B′′) after seven days of $esg^{ReDDM} > N^{DN}$. (**C–C′′**) RNAi-mediated downregulation of *Ecl* in $esg^{ReDDM} > N^{DN}$, forced expression of >*Ecl* (**D–D′′**) and >*EcR* RNAi (**E–E′′**) in $esg^{ReDDM} > N$ RNAi MF after seven days of tracing. Arrowheads highlight

*Figure 7 continued on next page*

*Figure 7 continued*

newly generated Dlg-1$^+$-EC (D+D''), scale bars = 25 µm. (**F**) Quantification of ISC progeny encompassing ISC-like, EC and EE classified after tumor size in R5 PMG (n = 13,5,7,4,6,9). Error bars are Standard Error of the Mean (SEM) and asterisks denote significances from one-way ANOVA with Bonferroni's Multiple Comparison Test (*p<0.05, **p<0.01; ***p<0,001; ****p<0.0001). (**G**) Quantification of newly generated EC (Dlg-1$^+$/RFP$^+$) in R5 PMG (n = 13,5,7,4,6,9). Error bars are Standard Error of the Mean (SEM) and asterisks denote significances from one-way ANOVA with Bonferroni's Multiple Comparison Test (*p<0.05, **p<0.01; ***p<0,001; ****p<0.0001). (**H**) Model of 20HE and JH hormonal pathways influencing physiological and pathological turnover in the intestine. Upon mating, JH from the neuroendocrine CA (*corpora allata*, [**Reiff et al., 2015**]) as well as ovarian 20HE (**a**) synergize on progenitor cells in the posterior midgut of adult female flies. The source of 20HE could not be definitely determined in this current study and 20HE might as well stem from the brain (**b**). JH induces ISC proliferation through Kr-h1, whereas 20HE signaling transduced by Eip75B-A /- C/PPARγ, ensures that newly produced EB differentiate into EC. New EC lead to a net increase in absorptive epithelium and thus ensures physiological adaptation of intestinal size to the new metabolic energy requirements of pregnancy. In the adult intestine, early steps of tumor-pathology are recapitulated when EC fate is inhibited by the lack of Notch signaling in progenitors (**Patel et al., 2015**). Notch mutant ISC divisions only produces ISC-like progenitors and EE. We show in this study that 20HE signaling, through Eip75B-A /- C/PPARγ, is capable to alleviate Notch tumor growth by driving intestinal progenitors into post-mitotic absorptive EC fate.

The online version of this article includes the following source data and figure supplement(s) for figure 7:

**Source data 1.** Data from *Figure 7*.

**Figure supplement 1.** Analysis of *Notch* mutant MARCM clones and consequences of clonal EcR-activation.

**Figure supplement 1—source data 1.** Data from *Figure 7—figure supplement 1*.

and *4*) as well as (iii) forced *EcI* expression induce proliferation and EB differentiation in MF mediated by *Eip75B-A /- C* (*Figures 2*, *3*, *6* and *7*). Together with the previous findings on JH upon mating (**Reiff et al., 2015**), our current data suggest a concerted role for JH on proliferation and 20HE on differentiation of midgut progenitors.

## The interplay between *Eip75B* and *Kr-h1* controls intestinal size adaptation

Downstream of the EcR, we found differential regulation of *Eip75B* isoforms upon mating. *Eip75B* activates egg production in ovarian GSC (**Ables and Drummond-Barbosa, 2010**; **König et al., 2011**; **Morris and Spradling, 2012**; **Uyehara and McKay, 2019**) and its knockdown reduces germarial size and number. In accordance with differentiation defects found in EB in our work, the number of cystoblasts, the immediate daughters of GSC, is reduced as well (**Ables and Drummond-Barbosa, 2010**; **Morris and Spradling, 2012**).

In our study, we dissected specific roles for *Eip75B* isoforms. *Eip75B-B* expression is at the lower detection limit (*Figure 3B*) and unchanged upon mating (*Figure 3C*). Thus, our finding that forced *Eip75B-B* expression raised ISC mitosis in VF (*Figure 3—figure supplement 1F*) might be due to ectopic expression. *Eip75B-B* mutants are viable and fertile and *Eip75B-B* is interacting with DNA only upon forming heterodimers with *Hormone receptor 3 (Hr3)* temporarily repressing gene expression (**Bialecki et al., 2002**; **Sullivan and Thummel, 2003**; **White et al., 1997**). Further studies of *Eip75B-B*, especially its expression pattern and transcriptional regulation, are necessary to elucidate in which physiological context, else than mating, *Eip75B-B* controls ISC proliferation.

Mating induction of *Eip75B-A /- C*, but not *Eip75B-B*, underlines their differential transcriptional regulation (**Segraves and Hogness, 1990**). This idea is supported by the finding that larval epidermis from *Manduca sexta* exposed to heightened JH-levels is sensitized to 20HE, which results in a 10-fold increase of *Eip75B-A*, but not of *Eip75B-B* expression (**Dubrovskaya et al., 2004**; **Zhou et al., 1998**). An in silico analysis of 4 kb regulatory regions upstream of *Kr-h1* using JASPAR revealed several EcR/USP and Met consensus sequences (**Khan et al., 2018**), suggesting that both hormonal inputs may also converge on *Kr-h1* driving ISC proliferation (*Figure 5B*; **Reiff et al., 2015**). We propose a synergistic interplay between JH- and 20HE-signaling upon mating, in which the JH pathway effector *Kr-h1* increases proliferation in ISC and the EcR effectors *Eip75B-A /- C* drive EB into differentiation. The combined action of *Kr-h1* and *Eip75B-A /- C* ensures mating induced organ size growth (*Figures 5K* and *7H*). The experimental investigation of *Eip75B* and *Kr-h1* regulatory regions incorporating endocrine signals from at least two different organs and leading to differential activity and function in ISC and EB will be a fascinating topic for future studies.

## *Eip75B/PPARγ* and progenitor fate commitment in physiology and pathology

A key discovery of our study is that systemic 20HE directs local intestinal progenitor differentiation. Upon EcR activation, we found *Eip75B-A /- C* promoting EB to EC differentiation. When tissue demand arises, EB are thought to physically separate from ISC and become independent of N. Detached EB acquire motility and are able to postpone their terminal epithelial integration after initial specification (*Antonello et al., 2015a*; *Antonello et al., 2015b*; *Martin et al., 2018*; *Siudeja et al., 2015*). Epithelial integration of progenitors can be initiated experimentally by *nubbin (nub)* mediated downregulation of *esg* and the nub-RB isoform is sufficient to initiate EC formation downstream of Notch signaling (*Korzelius et al., 2014*; *Tang et al., 2018*).

Our data suggest, that the ecdysone pathway acting through *Eip75B-A /- C* provides a similar, remotely inducible signal for terminal EB differentiation. This idea is supported by several lines of evidence: i) EB-specific knockdown of *Eip75B* stalls their differentiation using *klu^{ReDDM}* (*Figure 3—figure supplement 2E*), suggesting *Eip75B* functions after initial N-input. ii) *klu^+*-EB do not change fate upon *Eip75B* knockdown, suggesting a more mature EB differentiation status that already lost plasticity (*Figure 3—figure supplement 2A–D*; *Korzelius et al., 2019*; *Reiff et al., 2019*). iii) Forced *Eip75B-A /- C* expression in EB leads to their immediate differentiation independent of N-input (*Figure 6C,D*, *Figure 7—figure supplement 1D,E*). In addition, *Eip75B* was found to be highly expressed in EB in a recent study from the Perrimon lab (*Hung et al., 2018*), raising the question about its transcriptional regulation.

A plethora of studies established *Eip75B* as an early response gene downstream of the Ecdysone pathway (*Ables and Drummond-Barbosa, 2010*; *König et al., 2011*; *Morris and Spradling, 2012*; *Thummel, 1996*; *Uyehara and McKay, 2019*). Another pathway stimulating *Eip75B* expression in EB might be the N pathway. N activation leads to immediate EC differentiation of progenitors and thus phenocopies *Eip75B-A /- C* expression (*Hudry et al., 2016*; *Reiff et al., 2019*). Indeed, 20HE- and Notch-signaling have been shown to converge on target genes in various tissues and functions (*Mitchell et al., 2013*; *Sun et al., 2008*; *Xu et al., 2018*) and both pathways acetylate H3K56 modifying multiple regulatory genomic regions including *Eip75B* (*Skalska et al., 2015*). We observed that 20HE-signaling leads to the differentiation of progenitors to EC in a N LOF context (*Figures 5* and *6*, *Figure 7—figure supplement 1*). This is probably the result from strong genetic and pharmacological stimuli (*Figures 6* and *7*, *Figure 7—figure supplement 1*) acting through Eip75B. 20HE signaling might facilitate the expression of differentiation factors like *Eip75B* as well as *klu* in the presence of N signaling under normal physiological conditions (*Figure 7—figure supplement 1F*). Future in-depth analysis of *Eip75B* regulatory regions and the generation of *Eip75B* reporter flies will help to understand the interplay of 20HE and N signaling acting on *Eip75B*.

Confirming a close relationship between fly *Eip75B* and human *PPARγ*, we found midgut progenitors responding to the known PPARγ agonist Pioglitazone. In patients, PPARγ plays a role in i) pregnancy related adaptations of lipid metabolism (*Waite et al., 2000*) and ii) as target in colorectal cancer (CRC) (*Sarraf et al., 1999*). Decreased PPARγ levels are associated with development of insulin resistance and failure of lipolysis in obese pregnant women (*Rodriguez-Cuenca et al., 2012*; *Vivas et al., 2016*). Mating upregulates PPARγ/Eip75B (*Figure 3*) and we also describe increased lipid uptake upon EcR-activation (*Figure 2—figure supplement 1*) in absorptive EC. Srebp, a gene with known function in fatty acid metabolism, is also activated (*Horton et al., 2003*; *Reiff et al., 2015*; *Seegmiller et al., 2002*). Studying this interplay between SREBP, PPARγ and steroid hormone signaling will shed light on to which extent adaptation of intestinal size and lipid metabolism contribute to pathophysiological conditions like diabetes.

In CRC, PPARγ loss-of-function mutations are observed in around 8% of patients (*Sarraf et al., 1999*). PPARγ expression correlates with good prognosis and PPARγ is epigenetically silenced during CRC progression (*Ogino et al., 2009*; *Pancione et al., 2010*). In agreement with our in vivo experiments on Eip75B-A /- C in EB differentiation (*Figures 3–7*), in vitro studies identified *PPARγ* promoting differentiation in various colon cancer cell lines (*Cesario et al., 2006*; *Shimizu and Moriwaki, 2008*; *Yamazaki et al., 2007*; *Yoshizumi et al., 2004*).

In a CRC mouse model, biallelic loss of *PPARγ* leads to a 4-fold increase tumor incidence and reduced survival in female mice over males (*Apc^{Min/+}/PPARγ^{-/-}*) (*McAlpine et al., 2006*), whereas males develop around three times more colon tumors with wild type *PPARγ* (*Cooper et al., 2005*).

In line with this, ovariectomized $Apc^{Min/+}$ mice develop significantly more tumors and have decreased $PPAR\gamma$ expression, highlighting a possible link between $PPAR\gamma$ and estrogen signaling. Meta-analyses of hormone replacement studies have shown that estrogen confers a lower risk to develop CRC and better survival rates for female CRC patients (*Chen et al., 1998*; *Hendifar et al., 2009*; *Lin et al., 2012*).

It is tempting to speculate that this mechanism of steroid hormones signaling through EcR/ER (estrogen receptor) to activate *Eip75B/PPARγ* is conserved from flies to humans. Indeed, human estradiol and 20HE activate the EcR with similar affinity and elicit *Eip75B* transcription by binding to *EcRE* (Ecdysone-Responsive Elements). *EcRE* can be converted to functional *ERE* (estrogen responsive elements) by a simple change of nucleotide spacing suggesting high conservation (*Martinez et al., 1991*; *Schwedes et al., 2011*).

Taken together, our findings reveal that mating induced steroid hormone release, which signals to adjacent intestinal stem cells, controls their proliferation and, more importantly, differentiation of committed precursor cells through *Eip75B/PPARγ*. Ecdysone control of cell fate ensures the production of absorptive enterocytes during mating related intestinal adaptations and induces enterocyte fate in a *Drosophila* intestinal tumor model marked by loss of enterocyte differentiation. Mechanistically, we propose that *Eip75B/PPARγ* exerts an anti-neoplastic role by promoting progenitor differentiation into postmitotic enterocyte fate, thereby reducing the pool of mitotically active cells. Our findings might be a first step towards understanding the protective, but so far mechanistically unclear tumor suppressive role of steroid hormones in female colorectal cancer patients. Downstream of steroidal signaling, Eip75B/PPARγ promotes postmitotic cell fate when local signaling is deteriorated and thus might reflect a promising target for future studies in colorectal cancer models.

## Materials and methods

### Key resources table

| Reagent type (species) or resource | Designation | Source or reference | Identifiers | Additional information |
|---|---|---|---|---|
| Genetic reagent (*Drosophila melanogaster*) | $esg^{ReDDM}$ | *Antonello et al., 2015a* DOI: 10.15252/embj.201591517 | | *Figures 1–7; Figure 1—figure supplement 1; Figure 3—figure supplement 1* |
| Genetic reagent (*D. melanogaster*) | $esg^{ReDDM} > Eip75B\text{-}A$ | *Rabinovich et al., 2016* DOI: 10.1016/j.cell.2015.11.047 | | *Figure 3; Figure 3—figure supplement 1* |
| Genetic reagent (*D. melanogaster*) | $esg^{ReDDM} > Eip75B\text{-}B$ | *Rabinovich et al., 2016* DOI: 10.1016/j.cell.2015.11.047 | | *Figure 3; Figure 3—figure supplement 1* |
| Genetic reagent (*D. melanogaster*) | $esg^{ReDDM} > Eip75B\text{-}C$ | *Rabinovich et al., 2016* DOI: 10.1016/j.cell.2015.11.047 | | *Figure 3 ; Figure 3—figure supplement 1* |
| Genetic reagent (*D. melanogaster*) | $Srebp > CD8::GFP$ | *Reiff et al., 2015* DOI: 10.7554/eLife.06930 | | *Figure 2—figure supplement 1* |
| Genetic reagent (*D. melanogaster*) | $Mex >^{ts}$ | *Phillips and Thomas, 2006* DOI: 10.1242/jcs.02839 | | *Figure 2—figure supplement 1* |
| Genetic reagent (*D. melanogaster*) | MARCM (FRT2A) | *Lee and Luo, 1999* DOI: 10.1016/S0896-6273 (00)80701–1 | | *Figure 3—figure supplement 1* |
| Genetic reagent (*D. melanogaster*) | $Eip75^{A81}$-MARCM (FRT2A) | *Rabinovich et al., 2016* DOI: 10.1016/j.cell.2015.11.047 | | *Figure 3—figure supplement 1* |
| Genetic reagent (*D. melanogaster*) | $klu^{ReDDM}$ | *Reiff et al., 2019* DOI: 10.15252/embj.2018101346 | | *Figure 3—figure supplement 2* |
| Genetic reagent (*D. melanogaster*) | $N^{55e11}$-MARCM (FRT19A) | *Guo and Ohlstein, 2015* DOI: 10.1126/science.aab0988 | | *Figure 7—figure supplement 1* |
| Chemical compund, drug | RH5849 | DrEhrenstorfer | DRE-C16813000 | 340 µM final concentration |
| Chemical compund, drug | Pioglitazone | Sigma-Aldrich | Sigma-Aldrich 112529-15-4 | 14 µM final concentration |

## Genetics and fly husbandry/fly strains

*OregonR* and *w^1118* flies served as wild-type controls. The following transgenes and mutations were employed: *esg^ReDDM* (**Antonello et al., 2015a**), *klu^ReDDM* (**Reiff et al., 2019**), *UAS-EcI-Flag-HA* (**Okamoto et al., 2018**), *UAS-EcI* (**Okamoto et al., 2018**), *UAS-EcI-RNAi* (**Okamoto et al., 2018**), *UAS-Eip75B-A-Flag* (**Rabinovich et al., 2016**), *UAS-Eip75B-B-Flag* (**Rabinovich et al., 2016**), *UAS-Eip75B-C-Flag* (**Rabinovich et al., 2016**), *UAS-N^DN* (J. Treisman), *Dl::GFP* (F. Schweisguth), *N^55e11* *FRT19A* (**Guo and Ohlstein, 2015**), *Srebp-GAL4* (**Kunte et al., 2006**), *Mex-Gal4* (**Phillips and Thomas, 2006**), *Gbe+Su(H)dsRed* (T. Klein). From Bloomington *Drosophila* Stock Center (BDSC): *UAS-EcR-RNAi* (*BL58286*), *UAS-EcR.B2* (*BL4934*), *UAS-EcR.B2^W650A* (*BL9449*), *UAS-EcR.B2^F645A* (BL9450), *EcR^M554fs* (BL4894), *EcRE-lacZ* (BL4516), *Eip75B^A81*(BL23654), *UAS-CD8::GFP* (BL5137), *ovo^D1* (BL1309), *NRE::GFP* (BL30727), *NRE::GFP* (BL30728), *Dlg-1::GFP* (BL59417), From Kyoto *Drosophila* Stock Center: *UAS-Kr-h1* (DGRC120052). From FlyORF, Switzerland: *UAS-EcR-HA* (F000480), *UAS-Kr-h1* (F000495). From Vienna *Drosophila* Resource Center (VDRC) *UAS-EcR-RNAi (GD37059)*, *UAS-Eip75B-RNAi (GD44851)*, *UAS-Eip75B-RNAi (KK108399)*, *UAS-N-RNAi (GD14477)*.

## MARCM clones

Mosaic analysis with repressible cell marker (MARCM; *Lee and Luo, 1999*) clones were induced in midguts by Flippase under control of a *heat-shock* promoter. Expression of the Flippase was activated for 45 min in a 37°C-water bath to induce positively marked clones. Guts were dissected 5 days after induction. Clones in experimental and control flies were induced in parallel.

## Food composition and fly keeping

Fly food contained 1424 g corn meal, 900 g malt extract, 800 g sugar beet syrup, 336 g dried yeast, 190 g soy fluor, 100 g agarose, 90 ml propionic acid and 30 g NIPAGIN powder (antimycotic agent) in 20 l $H_2O$. Food was cooked for about an hour to reduce bioburden, then filled into small plastic vials and cooled down to RT. Flies were kept at 25°C except for crosses with temperature-sensitive GAL80^ts (GAL4 repressor) which were kept at 18°C (permissive temperature) until shifted to 29°C (restrictive temperature) to activate GAL4-mediated transgene expression. Crosses with *esg^ReDDM* and *klu^ReDDM* were carried out as described previously (**Antonello et al., 2015a**; **Reiff et al., 2015**; **Reiff et al., 2019**). Due to persisting problems with mucous formation on food surface in vials with VF, all experiments distinguishing between mated and virgin female flies were run on food with twice the amount of NIPAGIN. Mucous formation was avoided because of massive induction of tissue renewal by pathogenic stress.

## Hormone analogue treatments

A vial of fly food was reheated in the microwave until it became liquid, the hormone analogues were added, thoroughly mixed and filled into a new vial. For each ml of food 5 µl of RH5849 (340 µM final concentration; 20 µg/µl stock solution, diluted in MeOH; DRE-C16813000, DrEhrenstorfer) was added. As a control, an equivalent volume of carrier solution (MeOH) was added to the food. Hormone analogue treatments were performed for the period of the 7 days *ReDDM* (**Antonello et al., 2015a**) shift.

## PPARγ agonist treatments

A vial of fly food was reheated in the microwave until it became liquid, the PPARγ agonist was added, thoroughly mixed and filled into a new vial. For each ml of food 2.5 µl Pioglitazone (14 µM final concentration, 2 µg/µl stock solution, diluted in DMSO; Sigma-Aldrich, St. Louis, USA) were added. The equivalent amount of DMSO served as control. Flies were starved in an empty vial for at least six hours to ensure synchronized feeding when set to Pioglitazone. Food was renewed after three days and fly midguts were dissected after five days.

## Immunohistochemistry

Guts were dissected in PBS and transferred into 4% PFA immediately after dissection and staining was performed on an orbital shaker. After 45 min of fixation guts were washed once with PBS for 10 min. Antibodies were diluted in 0.5% PBT + 5% normal goat serum. The incubation with primary antibodies (1:250 anti-Dlg-1 [mouse; Developmental studies Hybridoma Bank (DSHB)]; 1:50 anti-Pros

[mouse; DSHB]; 1:2000 anti-pH3 [rabbit; Merck Millipore, 06–570]; 1:50 anti-EcR common Ag10.2 [mouse; DSHB]; 1:500 anti-HA High Affinity 3F10 [rat; Merck, Sigma-Aldrich]; 1:1500 anti-ß-Galacto-sidase preabsorbed [rabbit; Cappel Research Products]) was performed at 4°C over night. After washing with PBS guts were incubated with secondary antibodies (1:500 Goat anti-MouseAlexa647 [Invitrogen]; 1:500 Goat anti-RatAlexa647 [Invitrogen]; 1:500 Goat anti-RabbitAlexa647 [Invitrogen]) and DAPI (1:1000; 100 μg/ml stock solution in 0.18 M Tris pH 7.4; DAPI No. 18860, Serva, Heidel-berg) for at least 3 hr at RT. Guts were washed a last time with PBS prior to mounting in Fluoro-mount-G Mounting Medium (Electron Microscopy Sciences).

## X-Gal staining of *Drosophila* midguts

Guts were dissected in PBS and transferred into 4% PFA immediately after dissection. After 20 min of fixation, guts were washed three times with 0.3% PBT. Stainingbuffer (0.15M NaCl; 1 mM $MgCl_2$; 10 mM Na-phosphate buffer pH 7.2; 3.3 mM F3Fe(CN)6, 3.3 mM K4Fe(CN)6; 0.3% Triton X-100) was heated to 65°C and 3% X-Gal added. The guts were stained for at least 1 hr at 37°C until a dark blue staining became visible. Guts were washed two times in 0.3% PBT prior to mounting in Fluoro-mount-G Mounting Medium (Electron Microscopy Sciences). Stained midguts were imaged using an Axiophot2 microscope (Carl Zeiss) equipped with an AxioCam MRc (Carl Zeiss).

## OilRedO staining of *Drosophila* midguts

Guts were dissected in PBS and transferred into 4% PFA immediately after dissection. After 45 min of fixation, guts were washed in consecutive applications of 1xPBS, double-distilled $H_2O$, and 60% isopropanol. Guts were stained in a 6:4 dilution of OilRedO (Sigma-Aldrich, 0.1% stock solution diluted in isopropanol) in $dH_2O$ for 20 min, then washed in 60% isopropanol and $dH_2O$. After mount-ing in Fluoromount-G Mounting Medium (Electron Microscopy Sciences, emsdiasum) PMG were imaged using an Axiophot2 microscope (Carl Zeiss) equipped with an AxioCam MRc (Carl Zeiss). Oil-RedO staining intensity was analyzed using Fiji. RGB channels were split and the green channel was subtracted from the red channel to eliminate background signal. A fixed threshold was set, and guts were manually outlined as a ROI. The mean intensity of the resulting signal within the ROI was measured.

## Image acquisition

Posterior midguts were imaged using an LSM 880 Airyscan confocal microscope (Carl Zeiss) using 'Plan-Apochromat 10x/0.45 M27', 'Plan-Apochromat 20x/0.8 M27' and 'C-Apochromat 40x/1.20 W Corr M27' objectives. Image resolution was set to at least $2048 \times 2048$ pixels. At least three focal planes (1 μm distance) were combined into a Z-stack to image one cell layer and to compensate for gut curvature.

For determining whole midgut length, an Axio Zoom.V16 (Carl Zeiss) was employed with DAPI fil-ter and 1x/0.25-objective.

## Quantification of proliferation and intensity measurements

Maximum intensity projections were calculated from Z-stack images of PMG by Fiji (ImageJ 1.51 n, Wayne Rasband, National Institutes of Health, USA). Total cell number and RFP-positive cell count of *ReDDM* (*Antonello et al., 2015a*) guts were analyzed semi-automatically by a self-written macro for Fiji whereas GFP-positive cells were counted manually (macro available from the authors).

For fluorescence or OilRedO intensity measurements, intestines were scanned with fixed laser/exposure time settings and measured in Fiji. The region of interest was selected manually, and mean intensity of the area was determined. This way, relative EcR and ß-Galactosidase protein levels were measured in antibody-stainings, relative SREBP activity was analyzed in PMG cells expressing *mCD8::GFP* under the control of *Srebp-GAL4*, and amount of triglycerides was analyzed by OilRedO stainings.

## Jaspar

The open-access webtool JASPAR (*Khan et al., 2018*) was utilized to predict transcription factor binding sites within the 5'-UTR of *EcI*. It was specifically scanned for binding motifs related to Ecdy-sone and Juvenile Hormone signaling.

## RNA isolation and cDNA synthesis

The R5 regions or ovaries of at least three female flies were dissected and transferred into a droplet of RNA*later* Solution (Invitrogen by Thermo Fisher Scientific) on ice. The tissue was homogenized in 100 µl peqGOLD TriFast (VWR Life Science) and total RNA was isolated as specified by the manufacturer. The following cDNA synthesis was performed with 250 ng of total RNA and the SuperScript IV Reverse Transcriptase (Invitrogen by Thermo Fisher Scientific) using a 1:1 mixture of oligo-dT primers and random hexamers directly upon RNA isolation. Prior to Real-time qPCR, cDNA samples were diluted 1:2 in dH$_2$O.

## Verification of gene expression in the adult midgut by PCR

To verify gene expression in the adult *Drosophila* midgut, PCRs were performed with primer pairs specific for *ovo* and the *Svb* isoform, or isoform-specific primer pairs for *EcR* spanning at least one exon-exon boundary. PCRs were performed with Q5 High-Fidelity DNA Polymerase (NEB) for at least 30 cycles. Reaction products were loaded on an agarose gel (1.5%) and separated by electrophoresis to verify lengths of PCR products.

| Primer | Forward (5'—3') | Reverse (5'—3') |
|---|---|---|
| *EcR.A* | GGGGTCTAAGAAACATTTTGAGG | CCATTTGCAGCTGCAGCCGACGT |
| *EcR.B1* | GCACGTACGAAGCCCGATCGCGT | CCGGACTCGTTGCCGCAGAGCC |
| *EcR.B2* | GCACGTACGAAGCCCGATCGCGT | CTCTTCCCTCTGTTCACGCCC |
| *ovo* | CGCAGAGCCAAGATGTACGTG | GATAGTGGACCTCCGGCT |
| *Svb^Rep* | ACAGTAAGTTGCGAGCCGG | TGTTTTGGGGTGTCCTTTCGTG |

## Real-time qPCR

Expression levels of Ecdysone signaling pathway-associated genes were determined in VF and MF. Eclosed *OregonR* or *EcRE-LacZ* flies were aged for 4d before mating. After 72 hr of mating, RNA was isolated and cDNA synthetized before running qPCRs. After an enzyme activation step (20 s 95°C), 40 cycles of denaturation (2 s 95°C), primer annealing (20 s 58°C) and elongation (20 s 72°C) were run. Primers were designed to anneal at 59°C. Reaction was set up with KAPA SYBR FAST Universal (Roche) in a total volume of 10 µl. All qPCR results were normalized to the house-keeping gene *rp49*.

| Primer | Forward (5'—3') | Reverse (5'—3') |
|---|---|---|
| *Eip74EF-A* | AGAAACTTCGAGGCAATAGGGT | TGTGCGGCCTCATCTCAAG |
| *Eip74EF-B* | TGGCCATCCCACAACGC | GGGCGGAAATGAACCTGTTG |
| *Eip75B-A* | CCTGTGCCAGAAGTTCGATGA | AAGAATCCATCGGCATCTTCGT |
| *Eip75B-B* | CGTCTAGCTCGATTCCTGATCTA | CGGAAGAATCCCTTGCAACC |
| *Eip75B-C* | CTGTGGTTCCGGCGGATT | TCGAATTCTATGTTGAGTTCTGGTT |
| *EcI* | TGCAGTGCCGCTCTCAACTGTACC | TCACAGTAACCGTTGACCGCCTCC |
| *EcR.A* | GTGTTCGGTGAAAAACGCAA | TCCTAGCAACTGAGCTTTTGTAGAC |
| *EcR.B1* | TTAACGGTTGTTCGCTCGCA | AGTGCGGGAAACAATCAGAGCAT |
| *EcR.B2* | GTTAACGGTTGTTCGCTCGC | TGCGGGAAACAATCAGAGCATA |
| *Kr-h1(A)* | ACAATTTTATGATTCAGCCACAACC | GTTAGTGGAGGCGGAACCTG |
| *Kr-h1(B)* | AAATCTTGGGCACCCAAACAA | GTTGTGGCTGAATCTTTCGC |
| *Lac-Z* | ATCAGGATATGTGGCGGATGAGCG | AGTACAGCGCGGCTGAAATCATC |
| *rp49* | TGGTTTCCGGCAAGCTTCAA | TGTTGTCGATACCCTTGGGC |

## 20-HE isolation and titer determination

20-HE titers were determined in VF and MF of the same age. Eclosed *OregonR* or heterozygous *ovo*$^{D1}$ mutant flies were aged for 3d before mating. After 24 hr or 48 hr at least 20 adult female flies were pierced in the thorax with a needle and put into a 150 µl-PCR tube that was punctured in its very bottom. Hemolymph was harvested by centrifugation (5.000x g 5 min RT) and collected in a 1.5 ml-reaction tube. Total weight of flies was determined before and after centrifugation. Typical yields of hemolymph were around 0.6–1 mg. The isolated hemolymph was mixed with 500 µl MeOH, centrifuged (12.000x g 20 min 4°C) and the supernatant was transferred into a new 1.5 ml-reaction tube. MeOH was evaporated at 30°C in a vacuum centrifuge (Eppendorf concentrator plus) and 20HE was resuspended in 100 µl EIA buffer and stored at −20°C until usage.

Ecdysone levels were determined using the 20-Hydroxyecdysone Enzyme Immunoassay kit according to manufacturer's instructions (#A05120.96 wells; Bertin Bioreagent). 20HE titer was normalized to hemolymph yield.

## Statistical analysis

Figures of quantifications were assembled, and statistics were run in GraphPad Prism 6.01. For single comparisons, data sets were analyzed by two-sided unpaired t-test. For multiple comparisons, data sets were analyzed by one-way ANOVA and Tukey's post-hoc test. Significant differences are displayed as * for $p \leq 0.05$, ** for $p \leq 0.01$, *** for $p \leq 0.001$ and **** for $p \leq 0.0001$.

## Acknowledgements

We thank Maria Dominguez, Thomas Klein, Oren Schuldiner, Francois Schweisguth, Naoki Yamanaka, the Bloomington *Drosophila* Stock Center (NIHP400D018537), the Transgenic RNAi Project (TRiP) at Harvard Medical School (NIK/NIGMS R01-GM084947) and the Vienna *Drosophila* Resource Center (VDRC, http://www.vdrc.at) for providing transgenic fly stocks. We thank the Center for Advanced Imaging (CAi) at HHU for providing microscopy services. TR thanks Maria Dominguez in whose lab he initiated this project and Thomas Klein for being a very supportive host. We also thank Zeus Antonello, Nahuel Villegas, Hendrik Pannen and Thomas Klein for comments on the manuscript. The project is funded by a Deutsche Forschungsgesellschaft (DFG-Sachbeihilfe RE 34532–1) grant. LZ is supported by the Wilhelm Sander-Stiftung (2018.145.1).

## Additional information

### Funding

| Funder | Grant reference number | Author |
|---|---|---|
| Deutsche Forschungsgemeinschaft | RE 34532-1 | Tobias Reiff |
| Wilhelm Sander-Stiftung | 2018.145.1 | Lisa Zipper |

The funders had no role in study design, data collection and interpretation, or the decision to submit the work for publication.

### Author contributions

Lisa Zipper, Denise Jassmann, Sofie Burgmer, Bastian Görlich, Investigation; Tobias Reiff, Conceptualization, Funding acquisition, Investigation, Writing - original draft, Writing - review and editing

### Author ORCIDs

Tobias Reiff https://orcid.org/0000-0001-6610-6148

### Decision letter and Author response

Decision letter https://doi.org/10.7554/eLife.55795.sa1
Author response https://doi.org/10.7554/eLife.55795.sa2

## Additional files

### Supplementary files
• Transparent reporting form

### Data availability
All data generated or analysed during this study are included in the manuscript and supporting files. Source data files have been provided in a separate Excel File.

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
