## [Decision Letter]

**Decision letter after peer review:**

Thank you for submitting your article "Ecdysone steroid hormone remote controls intestinal stem cell fate decisions via the PPARγ-homologue E75B in *Drosophila*" for consideration by *eLife*. Your article has been reviewed by three peer reviewers, one of whom is a member of our Board of Reviewing Editors, and the evaluation has been overseen by a Reviewing Editor and K VijayRaghavan as the Senior Editor. The following individual involved in review of your submission has agreed to reveal their identity: Jerome Korzelius (Reviewer #3).

The reviewers have discussed the reviews with one another and the Reviewing Editor has drafted this decision to help you prepare a revised submission.

We would like to draw your attention to changes in our revision policy that we have made in response to COVID-19 (https://elifesciences.org/articles/57162). Specifically, when editors judge that a submitted work as a whole belongs in *eLife* but that some conclusions require a modest amount of additional new data, as they do with your paper, we are asking that the manuscript be revised to either limit claims to those supported by data in hand or to explicitly state that the relevant conclusions require additional supporting data.

Summary:

In this manuscript, the authors present evidence on the role of 20-Hydroxy-Ecdysone (20HE), secreted from the ovaries, in controlling proliferation and differentiation of intestinal stem cells into Enterocytes in the adult midgut of *Drosophila* females after mating. In the model presented, the authors propose that, following mating, 20HE secreted from the ovaries induce activation of EcR signaling within ISCs/EBs to drive EB differentiation into Enterocytes. They observed differential responses of 20HE on known target genes and followed the role of these target genes on stem cell proliferation and differentiation. The analysis of different isoforms of the ecdysone-target Eip75B reveals different roles for different isoforms of this PPARγ orthologue, with isoforms -A and -C being involved in regulation of EB-differentiation and the -B isoform inducing ISC mitosis. Additionally, activation of EcR signaling in Enterocytes is necessary for metabolic adaptation (lipid metabolism) in the gut in response to mating. Furthermore, the authors analysed the interplay between the 20HE- and the juvenile hormone-pathways. Interestingly, Ec signaling counteracts Notch driven tumourigenesis in the adult midgut.

Overall, this is an impressive volume of work that reveals a novel role for the ubiquitous ecdysone hormone in both ISC proliferation and progenitor differentiation. This connects well with recent work from the Miguel-Aliaga and Perrimon labs that has explored inter-organ communication from neighbouring tissues such as fat body and ovary to the intestine. It also invites further inquiry into the interplay between mating and the Insulin-dependent growth response upon re-feeding of the midgut (O'Brien et al., 2011).

Essential revisions:

According to the reviewers, there are two major conclusions that need to be revisited to be fully supported.

1) There is no evidence on the where the source of 20HE comes from. The authors claim that this is the ovary but there is no experimental proof of that. Suppression of 20HE synthesis in the ovary followed by assessment of ISC proliferation/differentiation and gut metabolic adaptation should address this point. Therefore, the systemic control of Ec signaling in the gut needs to be demonstrated. It represents the heart of the paper and what makes it more interesting/novel. In fact, towards the end of the abstract the authors conclude: 'To our knowledge, this is the first time a systemic hormone is shown to direct local stem cell fate decisions. ' This has not been tested and sources other than the ovaries could be playing a role.

2) The effect of Ec signaling activation on Notch-/- tumours does not prove a direct connection between the two phenomena. This could be the consequence of parallel action of two pathways with opposite effects. Is Ec signaling impaired in Notch tumours? This is key to connect both phenomena and substantiate a major claim of the paper.

The reviewers agreed that the Notch/Ec interaction is a bit overinterpreted in the manuscript, but some re-writing and combining all Notch results in Figure 5 and Figure 6 would fix this. The authors need to soften their claims and acknowledge the limitations of their experiments. This would also leave more room for some of the interesting data from the supplement regarding EE differentiation of E75A MARCM clones or the SREBP activation in Enterocytes to be included in the main manuscript.

Reviewer #1:

In this manuscript, the authors present data to show that in addition to juvenile hormone (shown in their previous paper, 2017) a second hormone, 20-Hydroxy-Ecdysone (20HE), controls proliferation and differentiation of intestinal stem cells in *Drosophila* females after mating. They used a system established previously, the ReDDM lineage method, to analyse the role of the 20HE-signalling pathway. They applied various genetic and pharmacological perturbations, such as overexpression and knock-down by RNAi and application of the ecdysone agonist, RH5849. They observed differential responses of 20HO on known target genes and followed the role of these target genes on stem cell proliferation and differentiation. They analysed the interplay between the 20HO- and the juvenile hormone-pathways. Finally, they present data to suggest that 20HO-mediated signalling can overcome the defects in differentiation observed upon inactivation of the Notch signalling pathway.

Major points:

1) Overall, these are interesting data and add novel insight into the way of tissue adaptation to external cues (here: mating). However, as presented here, the reader is supposed to have already detailed knowledge of the topic. In addition, this reader is often lost in details, with no easily understanding the essence. From this point of view, the manuscript would be better suited for a more specialised journal.

2) The text frequently uses the term "progenitor differentiation", based on the expression cell fate markers. Whether these cells differentiate to adopt their cell-type specific features (e. g. absorptive or endocrine) has not been studied. In addition, it is not clear what the authors mean when they state that "… results implicate EcR-signaling in.…. metabolic adaptation of EC" (last paragraph of Result section).

3) The authors demonstrate differential effects of 20HO-induced signaling on target gene expression, even on different splice variants of E75B. But how can a transcription factor (the EcR) affect the splicing of this target gene?

4) The text is often written sloppily, making it sometimes difficult to follow. One example: it reads: "After N specification, EB lineage is maintained.….". They probably mean that EBs have been specified by N signalling. The abbreviation N is not explained.

5) Figure 5 shows that overexpression of E75B-A/-C reduces the number of tumours induced by the absence of N. In addition, they "re-enable EC differentiation despite the lack of N". Again, this conclusion is based on the cell fate marker. It could still be that these cells do not differentiate properly at a later stage, they could even undergo apoptosis.

6) Figure 4: Quantification of all data shown in D-I would be helpful.

7) Figure 6H is not really clear.

8) In the Discussion section, some points were not addressed, e. g.

- What does it mean that E75B-B mRNA is not upregulated in mated females, but its overexpression induces ISC proliferation?

- How do the authors explain the shortening of the midgut, both upon E75B RNAi and E75B overexpression? Is the diameter enlarged?

Reviewer #2:

In this manuscript by Zipper et al., the authors present evidence on the role of Ecdysone (Ec/20HE) signaling regulating ISC proliferation and EB differentiation in the adult *Drosophila* midgut. In their model, the authors propose that, following mating, 20HE secreted from the ovaries induce activation of EcR signaling within ISCs/EBs to drive EB differentiation into Enterocytes. Additionally, activation of EcR signaling in Enterocytes is necessary for metabolic adaptation (lipid metabolism) in the gut in response to mating. Interestingly, Ec signaling counteracts Notch driven tumourigenesis in the adult midgut. The work in this report complements a previous publication from the senior author on the role of systemic juvenile hormone influencing proliferation and metabolic adaptation of the female gut in response to reproductive demands. There are, however, inconsistencies in some of the data and key aspects of the model that need more direct demonstration through experiments.

1) While the role of EcR signaling activation hormone import in the gut is reasonably demonstrated, there is no evidence on the where the source of 20HE comes from. The authors claim that this is the ovary but there is no experimental proof of that. Suppression of 20HE synthesis in the ovary followed by assessment of ISC proliferation/differentiation and gut metabolic adaptation should address this point.

2) Related to the point above: ISCs proliferation is inferred as the number of esg >gfp, rfp+ve cells throughout most of the paper. This should be directly measured by looking at pH3 staining.

3) Similarly, current data on changes in lipid metabolism (Figure S2C-G) should be complemented with direct assessment of gut lipid content.

4) Are the metabolic and ISC phenotypes influenced by EcR signaling linked? What happens to ISC proliferation if Ec import to the Enterocytes is blocked?

5) The use of SREBP-gal4 to manipulate gut Enterocytes (Figure S2—figure supplement 1E, F) is unconventional. Mex-gal4 or MyoIA-gal4, should be used.

6) Images should be presented for the data in Figure 2H.

7) The phenotype resulting from overexpression of E75B-A and E75B-C appears much stronger than that form EcI overexpression of the use of the EcR agonist. How do the authors explain that?

8) Genetic interactions presented in Figure 4 should be quantified (ISC proliferation and differentiation) and controls with individual gene manipulations included.

9) Can the phenotype of EcI overexpression or EcR activation be modified by knocking down E75B? Reciprocally, can E75B-A or -C overexpression rescue EcI knockdown?

10) The phenotype of E75B-A MARCM clones is unclear. A time course should be done. The authors claim Enteroendocrine cells are lost. However, I cannot see that in the data presented. This should be quantified. EE cell stainings are overall unclear.

11) The effect of Ec signaling activation on Notch-/- tumours does not prove a direct connection between the two phenomena. This could be the consequence of parallel action of two pathways with opposite effects. Is Ec signaling impaired in Notch tumours? This is key to connect both phenomena and substantiate a major claim of the paper.

Reviewer #3:

This work describes the role of ecdysone signaling, especially its downstream target Eip75 in midgut homeostasis in *Drosophila*. As ecdysone spikes after mating, and the intestine was already shown to be receptive to JH-signaling from the ovary in previous work by the last author, the authors start with addressing the role of the ecdysone receptor EcR in ISC/EB proliferation and differentiation. Next, the role of the ecdysone-target Eip75B is explored. Interestingly, the analysis of different isoforms reveals different roles for different isoforms of this PPAR-γ orthologue, with isoforms -A and -C being involved in regulation of EB-differentiation and the -B isoform inducing ISC mitosis. The interaction with JH-signaling is investigated and lastly, the role of EcR and its downstream target Eip75B is assessed in Notch LOF-tumors. Altogether, this is an impressive volume of work that reveals a new role for the ubiquitous ecdysone hormone in both ISC proliferation and progenitor differentiation. This connects well with recent work from the Miguel-Aliaga and Perrimon labs that has explored inter-organ communication from neighbouring tissues such as fat body and ovary to the intestine. It also invites further inquiry into the interplay between mating and the Insulin-dependent growth response upon re-feeding of the midgut (O'Brien et al., 2011). I would recommend this work for publication after the issues described below would be addressed. Especially in the light of the current Covid19-crisis, I do not think extra experimental work should be necessary.

Major comments:

1) Central to their thesis is the fact that ecdysone produced in the ovary would signal to ISCs/EBs to exert its effect. However, the authors have not done any genetic experiments to show that flies without ecdysone production in the ovary would not have the mating-induced response ISC-EB increase. Hence, a role of local ecdysone production by either the fat body or gut cannot be ruled out. Related to this, the authors decribe determination of 20HE-titers on wild-type and ovoD1 heterozygotes in their Materials and methods section, but no data can be found in the manuscript. Can the authors comment on that and include the 20HE-titer data?

2) This relates more to the parallels drawn between the work described here and the parallels with mammalian steroid hormone signaling. Based on their results, I do not think the statement that "our findings suggest a tumor-suppressive role for steroidal signalling " is warranted. Loss of EcR suppresses ISC/EB proliferation upon mating and the EcR agonist RH5849 dramatically induces proliferation, which would be more in line with an oncogenic effect of EcR-signaling in the fly. I understand that EcR induces Eip75, a PPARγ orthologue and they provide evidence that this induces differentiation and therefore impairs Notch LOF-driven tumorigenesis, in line with mammalian data. I would suggest altering some of these statements and moving them out of the Abstract and Introduction to the Discussion section, where there is more room to elaborate on these contradictions.

---

## [Author Response]

Summary:In this manuscript, the authors present evidence on the role of 20-Hydroxy-Ecdysone (20HE), secreted from the ovaries, in controlling proliferation and differentiation of intestinal stem cells into Enterocytes in the adult midgut of *Drosophila* females after mating. In the model presented, the authors propose that, following mating, 20HE secreted from the ovaries induce activation of EcR signaling within ISCs/EBs to drive EB differentiation into Enterocytes. They observed differential responses of 20HE on known target genes and followed the role of these target genes on stem cell proliferation and differentiation. The analysis of different isoforms of the ecdysone-target Eip75B reveals different roles for different isoforms of this PPARγ orthologue, with isoforms -A and -C being involved in regulation of EB-differentiation and the -B isoform inducing ISC mitosis. Additionally, activation of EcR signaling in Enterocytes is necessary for metabolic adaptation (lipid metabolism) in the gut in response to mating. Furthermore, the authors analysed the interplay between the 20HE- and the juvenile hormone-pathways. Interestingly, Ec signaling counteracts Notch driven tumourigenesis in the adult midgut.Overall, this is an impressive volume of work that reveals a novel role for the ubiquitous ecdysone hormone in both ISC proliferation and progenitor differentiation. This connects well with recent work from the Miguel-Aliaga and Perrimon labs that has explored inter-organ communication from neighbouring tissues such as fat body and ovary to the intestine. It also invites further inquiry into the interplay between mating and the Insulin-dependent growth response upon re-feeding of the midgut (O'Brien et al., 2011).Essential revisions:According to the reviewers, there are two major conclusions that need to be revisited to be fully supported.1) There is no evidence on the where the source of 20HE comes from. The authors claim that this is the ovary but there is no experimental proof of that. Suppression of 20HE synthesis in the ovary followed by assessment of ISC proliferation/differentiation and gut metabolic adaptation should address this point. Therefore, the systemic control of Ec signaling in the gut needs to be demonstrated. It represents the heart of the paper and what makes it more interesting/novel. In fact, towards the end of the abstract the authors conclude: 'To our knowledge, this is the first time a systemic hormone is shown to direct local stem cell fate decisions. ' This has not been tested and sources other than the ovaries could be playing a role.

We fully agree that the 20HE source and determining 20HE titers were important experiments to perform. 20HE titer measurements in ovaries and hemolymph of VF vs MF are part of Figure 2A-C now. As reviewer 3 commented, we began measuring 20HE titers in wild-type and *ovo^D1^* female flies. Unfortunately, due to the Covid19 crisis, we waited more than 5 months for a last immunoassay Kit to arrive to finally complete three biological replicas. The kit arrived finally two months after the initial submission. The corresponding new data and text were added to:

Results section: “In close anatomical proximity to the PMG, the ovaries are an established ecdysteroidogenic tissue. Determining 20HE titers 48hours after mating using enzyme immunoassays, we confirmed previous reports of mating dependent increases of ovarian 20HE titers (Figure 2A) (Ameku and Niwa, 2016, Harshman et al., 1999) and observed a similar increase in the hemolymph (Figure 2A). As a study investigating the ecdysoneless mutants suggested that the ovary is the only source for 20HE in adult females (Garen et al., 1977), we sought to diminish 20HE titers in adult females. Therefore, we genetically ablated the ovaries using the dominant sterile ovo^D1^ allele in which egg production is blocked prior to vitellogenesis (Busson et al., 1983, Oliver et al., 1987, Reiff et al., 2015) (Figure 2D). 20HE titers in the hemolymph of ovo^D1^ females are reduced around 40-50% compared to wild-type females (Figure 2B,A). Interestingly, 20HE titers in hemolymph and remnants of the ovaries are still significantly increased upon mating of ovo^D1^ females (Figure 2C). Consequently, 20HE titers in sterile esg^ReDDM^/ovo^D1^ MF increase the number of progenitors (Figure 2E) and progeny (Figure 2F). This suggests that remaining 20HE levels are sufficient to elicit mating related midgut adaptations.”

Discussion section: “Our data supports the idea that 20HE and JH synergistically concert intestinal adaptations balancing nutrient uptake to increased energy demands upon mating (Figure 1, Figure 2, Figure 3, Figure 4, Figure 5). We confirmed that mating induces ovarian ecdysteroid biosynthesis stimulating egg production by ovarian GSC (Figure 2A)(Ables and Drummond-Barbosa, 2010, Ameku and Niwa, 2016, Ameku et al., 2017, Harshman et al., 1999, König et al., 2011, Morris and Spradling, 2012, Uyehara and McKay, 2019). In accordance with an interorgan signaling role for 20HE, we detected increased 20HE titers in the hemolymph (Figure 2A). Surprisingly, we found this mating dependent increase of 20HE titers still present upon partial genetic ablation of the ovaries using ovo^D1^ (Figure 2B)(Reiff et al., 2015). This can be explained by either (i) other source(s) for 20HE in adult females like the brain (Chen et al., 2014, Itoh et al., 2011) or (ii) 20HE release from remaining germarial cells in stage 1-4 eggs of ovo^D1^ VF and MF. Indeed, germarial cells have been shown to express the ecdysteroidogenic Halloween genes (Ameku and Niwa, 2016, Ameku et al., 2017). In addition, ovo^D1^ MF lack ovarian 20HE uptake during vitellogenesis (Figure 2D) of later egg stages, which potentially contributes to 20HE titer in the hemolymph of ovo^D1^ females and intestinal epithelium expansion (Figure 2D,E,J,K) (Enya et al., 2014).

The epithelial expansion upon mating of ovo^D1^ females has been observed before (Reiff et al., 2015) and might be comparably high due to the different genetic background of ovo^D1^ flies compared to *w1118* (Figure 2Q). Taken together, our current experiments cannot clearly dissect whether the ovary is the exclusive source of 20HE (Garen et al., 1977). In future experiments, complete ablation of ovarian 20HE by genetically removing GSC (Flatt et al., 2008, Kai and Spradling, 2003) or from other sources like the brain need to be performed to address the exact source(s) (Chen et al., 2014, Itoh et al., 2011).”

We also considered that *ovo^D1^* might act directly on intestinal homeostasis due to the induction of EC production (Figure 2Q). A current preprint from Francois Payre and Dani Osman suggests that the EcR induced post translational modification of somatically expressed *shavenbaby (svb^Rep^)* from the *ovo* gene locus controls ISC self-renewal and EC differentiation (Al Hayek et al., 2019). (Consistent with our study, they also observe less progenitors expressing EcR-RNAi and a dominant-negative EcR construct). After consulting the senior authors, a role for *ovo^D1^* acting on somatic *shavenbaby* can be excluded (Francois Payre, University of Toulouse, personal communication) (Andrews et al., 1998, Delon et al., 2003, Mevel-Ninio et al., 1996).

Taken together, we cannot clearly dissect with these experiments whether the ovary (or the remaining cells after *ovo^D1^* ablation) are the exclusive source of 20HE. We now reworked the manuscript accordingly and discuss other sources than the ovaries.

2) The effect of Ec signaling activation on Notch-/- tumours does not prove a direct connection between the two phenomena. This could be the consequence of parallel action of two pathways with opposite effects. Is Ec signaling impaired in Notch tumours? This is key to connect both phenomena and substantiate a major claim of the paper.

The reviewers raise an important and interesting issue. Downstream of the 20HE signaling, we found differential regulation of *Eip75B* isoforms and to investigate their timely expression and function is of high importance, therefore we aimed to tackle them experimentally.

As we discuss in the revised manuscript now, Eip75B might not only be a ‘classical’ effector of 20HE signaling but also a target of Notch activation. Convergence of EcR and Notch signaling are known from various tissues and functions. Acting together, EcR and Notch were shown to regulate processes like proliferation, differentiation and endocycling/gene amplification by converging on transcription factors like *cut*, *tramtrack* and the broad complex (Mitchell et al., 2013, Sun et al., 2008, Xu et al., 2018).

A possibility how both pathways may converge on *Eip75B* is indicated by a study from Sarah Brays lab. The study suggests that active EcR and Notch, via H3K56 histone acetylation, modify multiple regulatory regions including *Eip75B* (Skalska et al., 2015). A thorough investigation of *Eip75B* regulatory regions is required to elucidate in detail how both pathways may affect *Eip75B*.

Experimentally, we tried to tackle 20HE signaling inside of Notch tumors during the revision process. We followed several approaches:

We investigated EcR activity in Notch tumors by crossing *esg^ReDDM^* flies with *EcRE-LacZ* and *>N-RNAi*. After X-Gal stainings failed on EcRE-LacZ PMG (reviewer#2, point6), we performed (i) immunohistochemistry, (ii) qPCR and iii) activating EcR in Notch activity reporter flies:

i) We used antibodies against the *EcRE-LacZ* gene product β-Galactosidase and compared intra- and extra tumoral EcRE signal intensity (A-D). We found a decrease of β-Galactosidase signal intensity in Notch tumors compared to control ISC/EB (-23,2%, D) and surrounding EC (-22,7%, D). Interestingly this reduction in EcRE-LacZ intensity was restricted to the ISC fraction (B) and no difference was observed in the EE fraction of N-tumors (C).

However, there is conflicting data about the intracellular localization of the EcRE-lacZ gene product in the literature (Kozlova and Thummel 2003, Neto et al., 2017, Okamoto et al., 2018, Schwedes et al., 2011). Okamoto et al., clearly show an enrichment of signal in the nucleus, whereas the original paper from Kozlova, the Schwedes and Neto publications and our images suggest cytoplasmatic localization of the β-Gal enzyme (A-C). Given this discrepancy in the literature, we decided not to include this finding in the main manuscript and state that 20HE signaling is affected in N-tumors. However, we are open to the reviewer’s expertise and opinion about including it or not.

ii) We addressed EcRE-LacZ activity with qPCR in guts carrying Notch tumors (*esg^ReDDM^/>N-RNAi*, E), but found no significant change compared to control guts (*esg^ReDDM^/+*, E). However this experiment is difficult in its interpretation, as whole gut cDNA is used and changes in the most abundant non-manipulated cells (EC) might occur when N-tumors are induced (Patel et al. 2015). Similar intricacies are true for qPCR for *Eip75B* isoform expression levels in (F), in which we also did not detect any difference in N-tumor (*esg^ReDDM^/>N-RNAi*) vs. control guts (*esg^ReDDM^/+*) after seven days of tumor induction. FACS sorting of esg^+^-cells for those experiments would sufficiently enrich the ISC/EB fraction to perform these experiments in a more convincing way, but is not readily established.

iii) In a third experiment, we activated EcR-signaling in Gbe-SuH reporter flies (reviewer 2, point 10) using RH5849 and detected a significant increase in N-activity (G). This experiment shows that under physiological conditions, N activity is indeed impacted by EcR-signaling leading to EC generation (Figure 2I, Figure 7—figure supplement 1F). Thus, our data suggests that *Eip75B* elicits EC differentiation downstream of EcR- and Notch activation, and thus may incorporate signals from both pathways (Figure 3, Figure 4, Figure 6, Figure 7, Figure 3—figure supplement 1, Figure 3—figure supplement 2 Figure 7—figure supplement 1 and reviewer 2, point 10). This interesting interplay on *Eip75B* and also the role of the Notch target gene *klu* warrant future studies (Korzelius et al., 2019, Reiff et al., 2019).

Again, we thank the reviewer(s) for this important comment and adjusted our claims and acknowledged the limitations of our N LOF experiments in the revised manuscript. We added Figure 6—figure supplement 1F to the revised manuscript and now dedicate a paragraph addressing the possible regulation of *Eip75B* by both pathways in the Discussion.

**Author response image 1. sa2fig1:** 

The reviewers agreed that the Notch/Ec interaction is a bit overinterpreted in the manuscript, but some re-writing and combining all Notch results in Figure 5 and Figure 6 would fix this. The authors need to soften their claims and acknowledge the limitations of their experiments. This would also leave more room for some of the interesting data from the supplement regarding EE differentiation of E75A MARCM clones or the SREBP activation in Enterocytes to be included in the main manuscript.

We agree and adjusted the claims of our data concerning Notch and 20HE. We re-wrote various paragraphs in the revised manuscript. For details, please see the individual reviews.

Reviewer #1:In this manuscript, the authors present data to show that in addition to juvenile hormone (shown in their previous paper, 2017) a second hormone, 20-Hydroxy-Ecdysone (20HE), controls proliferation and differentiation of intestinal stem cells in *Drosophila* females after mating. They used a system established previously, the ReDDM lineage method, to analyse the role of the 20HE-signalling pathway. They applied various genetic and pharmacological perturbations, such as overexpression and knock-down by RNAi and application of the ecdysone agonist, RH5849. They observed differential responses of 20HO on known target genes and followed the role of these target genes on stem cell proliferation and differentiation. They analysed the interplay between the 20HO- and the juvenile hormone-pathways. Finally, they present data to suggest that 20HO-mediated signalling can overcome the defects in differentiation observed upon inactivation of the Notch signalling pathway.Major points:1) Overall, these are interesting data and add novel insight into the way of tissue adaptation to external cues (here: mating). However, as presented here, the reader is supposed to have already detailed knowledge of the topic. In addition, this reader is often lost in details, with no easily understanding the essence. From this point of view, the manuscript would be better suited for a more specialised journal.

We thank the reviewer for pointing this out and we have revised our manuscript addressing the reviewer´s concerns. The revised manuscript now aims to underline the more general implications of our findings for the broad readership of *eLife*. As an example, we now include data showing that the PPARγ agonist Pioglitazone acts through Eip75B (Figure 4) that points to a more general relevance of our findings. Regarding the interest for *eLife* readers: We think that for stem cell biologists the demonstration how a hormone acts through Eip75B on progenitor fate decisions is of considerable interest. The *Drosophila* research community profits from the understanding of synergistic action of hormones acting sequentially on different cell types in the context of mating. Finally, for clinical researchers the defined in vivo role of *Eip75B/PPARγ* in intestinal progenitors might be of interest in the future as a plethora of activating and inhibiting ligands is readily available. Our data highlights intestinal progenitor differentiation as a new and important aspect of how hormones are involved in physiological and pathological intestinal homeostasis.

2) The text frequently uses the term "progenitor differentiation", based on the expression cell fate markers. Whether these cells differentiate to adopt their cell-type specific features (e. g. absorptive or endocrine) has not been studied. In addition, it is not clear what the authors mean when they state that "… results implicate EcR-signaling in.…. metabolic adaptation of EC" (last paragraph of Result section).

The ReDDM method combined with esg-Gal4 allows to distinguish between the ISC/EB progenitor cell types (GFP+/RFP+) and the differentiated progeny (GFP-/RFP+, EE and EC, Antonello et al., 2015). Our term RFP-only progeny from the quantifications encompasses terminally differentiated absorptive EC marked by Discs-large-1 (Dlg-1 marks smooth septate junctions in highly polarized cells integrated into the epithelium (Chen et al., 2018a, Furuse and Izumi, 2017, Izumi et al., 2019, Izumi et al., 2016, Izumi et al., 2012)) and EE marked by prospero (Ohlstein and Spradling, 2006). We and other labs did not observe any other differentiated or non-differentiated RFP^+^-only cell type stemming from *esg^ReDDM^* tracings (Al Hayek et al., 2019, Arthurton et al., 2019, Martin et al., 2018, Mundorf et al., 2019, von Frieling et al., 2020).

We now include new supplementing quantification for data in Figure 3 showing that the abundance of EE fate in Eip75B manipulations is less than one new EE (Figure 3—figure supplement 1E) of more than 300 RFP+-only counts (Figure 3M). In various other figures (Figure 5, Figure 6, Figure 7, Figure 3—figure supplement 2 and Figure 7—figure supplement 1), we present images co-stained with terminal differentiation markers Dlg-1 and Pros. For figures in which cell fate changes were investigated, we also now provide quantifications of Dlg-1+/RFP+ upon activated Eip75B (Figure 4F) and 20HE signaling (Figure 6I, Figure 7—figure supplement 1C, Figure 7G) and the induction of EC fate.

Concerning the metabolic adaptation of EC (Figure 2—figure supplement 1C-O), we were referring to the upregulation of Srebp upon mating, which is accompanied by the upregulation of transcripts of fatty acid synthesis (long-chain fatty acid CoA ligases bubblegum (bgm), AcylCoA synthetase long-chain (Acsl), Fatty acid synthase (FAS) and Acetyl-CoA carboxylase (ACC)) described in (Reiff et al. 2015). We additionally provide data now showing RH5849 and *>EcI* raising lipid uptake by directly addressing lipid uptake with OilRedO stainings (Figure 2—figure supplement 1I-O). We added and rephrased the revised manuscript accordingly.

3) The authors demonstrate differential effects of 20HO-induced signaling on target gene expression, even on different splice variants of E75B. But how can a transcription factor (the EcR) affect the splicing of this target gene?

The reviewer is right and we replaced the term ‘splice variant’ with protein ‘isoform’ throughout the manuscript as there is no evidence for alternative splicing of Eip75B. However, it cannot be excluded that such regulation exists for Eip75B gene products as it was recently shown that e.g. JH primes the ecdysteroid response through alternative splicing of *taiman* in mosquitoes (Liu et al., 2018).

4) The text is often written sloppily, making it sometimes difficult to follow. One example: it reads: "After N specification, EB lineage is maintained.….". They probably mean that EBs have been specified by N signalling. The abbreviation N is not explained.

We carefully proofread the manuscript again and corrected many phrases that were long and thus difficult to understand. The abbreviation Notch (N) is correctly introduced in the second paragraph of the Introduction. In addition, we prepared a list of abbreviations following the reviewer’s suggestion.

5) Figure 5 shows that overexpression of E75B-A/-C reduces the number of tumours induced by the absence of N. In addition, they "re-enable EC differentiation despite the lack of N". Again, this conclusion is based on the cell fate marker. It could still be that these cells do not differentiate properly at a later stage, they could even undergo apoptosis.

The cell fate marker (Dlg-1+/RFP+ EC) in new Figure 6 and Figure 7 faithfully reflects terminal differentiation (see reviewer 1, point 2 for references and details).

There are several lines of evidence arguing against a role for PCD in EB survival. (i) We directly addressed programmed cell death with cleaved-caspase 3 feeding flies with RH5849 (mimicking 20HE mating induction) and found no EB death. (ii) PCD in EB is comparably easy to detect by membrane-blebbing and irregularities (GFP+-vesicles) when ReDDM tracing is used (Reiff et al., 2019). (iii) A later PCD as fully differentiated pros^+^-EE and Dlg-1^+^-EC (see reviewer 1, point 2) is unlikely as transgenes are not *esg*- or *klu-Gal4* activated anymore in differentiate cell types (Figures 1A-B, Figure 3—figure supplement 2A). (iv) Upon Eip75B-A/-C expression, ISC/EB progenitor numbers are strongly reduced (Figure 3L). The high number of differentiated cells upon Eip75B-A/-C expression (Figure 3M) argues against a significant contribution of EB-death, but rather shows EB to EC differentiation (inset Figure 3G,H,K) as the main cause of progenitor compartment reduction. Taken together our data argues against a major role of cell loss in neither progenitor- nor differentiated progeny populations.

6) Figure 4: Quantification of all data shown in D-I would be helpful.

In the time after the submission and during the revision, we were able to raise the replica numbers of these genotypes and perform a statistical analysis of all genotypes now. We added this information to new Figure 5 now and incorporated it into the revised manuscript.

7) Figure 6H is not really clear.

New Figure 7H is providing a model of our current and published findings about hormonal control of midgut progenitors in physiology and pathology. We extended and specified its description in the figure legend now.

8) In the Discussion section, some points were not addressed, e. g.- What does it mean that E75B-B mRNA is not upregulated in mated females, but its overexpression induces ISC proliferation?

The topic of Eip75B-B regulation is indeed interesting. We already discussed *Eip75B-B* function and regulation in subsection “The interplay between *Eip75B* and *Kr-h1* controls intestinal size adaptation”. “In our study, we dissected specific roles for Eip75B isoforms. Eip75B-B expression is at the lower detection limit (Figure 3B) and unchanged upon mating (Figure 3C). Thus, our finding that forced Eip75B-B expression raised ISC mitosis in VF (Figure 3—figure supplement 1F) might be due to ectopic expression. Eip75B-B mutants are viable and fertile and Eip75B-B is interacting with DNA only upon forming heterodimers with Hormone receptor 3 (Hr3) temporarily repressing gene expression (Bialecki et al., 2002, Sullivan and Thummel, 2003, White et al., 1997). Further studies of Eip75B-B, especially its expression pattern and transcriptional regulation, are necessary to elucidate in which physiological context, else than mating, Eip75B-B controls ISC proliferation.”

One has to keep in mind that Eip75B-B expression in ISC with UAS constructs is forced and could be ectopic. *Eip75B-B* levels on whole midgut cDNA are low (Figure 3B) and could originate from another cell type than progenitors. The lack of a Zn-finger domain in Eip75B-B and the interaction with Hr3 definitely warrants an in-depth future investigation. We reworked the manuscript concerning this point and highlight now that the stimulus leading to Eip75B-B is unknown (Figure 5K).

- How do the authors explain the shortening of the midgut, both upon E75B RNAi and E75B overexpression? Is the diameter enlarged?

We reworked this part of the Results section now and added some details. The text goes as follows now: “Disrupting intestinal homeostasis alters midgut length (Hudry et al., 2016) and in line with this, Eip75B manipulations result in a shorter midgut as EC make up around 90% of the whole midgut epithelium (Figure 3—figure supplement 1G). Eip75B affects intestinal homeostasis by either slowing down terminal EB to EC differentiation (Eip75B-B and Eip75B-RNAi) or depleting the ISC and EB pool (Eip75B-A/–C, Figure 3—figure supplement 1G) preventing sufficient EC production along the gut. When EC production is blocked using >N-RNAi, the midgut shrinks around one third over a period of seven days (Chen et al., 2018b, Guo and Ohlstein, 2015, Micchelli and Perrimon, 2006, Ohlstein and Spradling, 2006, Ohlstein and Spradling, 2007, Patel et al., 2015). Thus, all Eip75B manipulations ultimately lead to an insufficient number of new absorptive EC resulting in a shorter midgut.”

We also added measurements of midguts with Notch-tumors combined with *esg^ReDDM^* to illustrate how much total block of EC production reduces midgut length (~1/3, Figure 3—figure supplement 1G). Concerning midgut diameter measurements: we found the diameter to fluctuate more than midgut length and also depend on different food recipes. We found measuring midgut length resulting in more robust values in concordance with the work of Bruno Hudry and Irene Miguel-Aliaga (Hudry et al., 2016).

Reviewer #2:In this manuscript by Zipper et al., the authors present evidence on the role of Ecdysone (Ec/20HE) signaling regulating ISC proliferation and EB differentiation in the adult *Drosophila* midgut. In their model, the authors propose that, following mating, 20HE secreted from the ovaries induce activation of EcR signaling within ISCs/EBs to drive EB differentiation into Enterocytes. Additionally, activation of EcR signaling in Enterocytes is necessary for metabolic adaptation (lipid metabolism) in the gut in response to mating. Interestingly, Ec signaling counteracts Notch driven tumourigenesis in the adult midgut. The work in this report complements a previous publication from the senior author on the role of systemic juvenile hormone influencing proliferation and metabolic adaptation of the female gut in response to reproductive demands. There are, however, inconsistencies in some of the data and key aspects of the model that need more direct demonstration through experiments.1) While the role of EcR signaling activation hormone import in the gut is reasonably demonstrated, there is no evidence on the where the source of 20HE comes from. The authors claim that this is the ovary but there is no experimental proof of that. Suppression of 20HE synthesis in the ovary followed by assessment of ISC proliferation/differentiation and gut metabolic adaptation should address this point.

We thank the reviewer for this experimental suggestion. Please find new data and discussion in Essential revisions 1.

2) Related to the point above: ISCs proliferation is inferred as the number of esg >gfp, rfp+ve cells throughout most of the paper. This should be directly measured by looking at pH3 staining.

We agree with the reviewer that momentary ISC proliferation is best addressed with pH3 or comparable markers of proliferation. However, concerning questions related to homeostatic turnover of the intestinal tissue, cell tracing systems like MARCM clones (Lee and Luo, 2001) and later Bruce Edgars esg-FlpOut (Jiang et al., 2009) and our ReDDM method (Antonello et al., 2015), have been proven highly reliable to address stem cell production over time. One also has to keep in mind that additional regulatory mechanisms like nutrition, delayed differentiation and even EB death strongly affect homeostatic processes and thus momentary ISC proliferation as well (Antonello et al., 2015, O'Brien et al., 2011, Reiff et al., 2019, Reiff et al., 2015).

Additionally, ISC proliferation was shown to strongly depend on the circadian rhythm (Karpowicz et al., 2013), thus a ‘snapshot‘ of ISC proliferation using pH3 after seven days can be even misleading when overall tissue turnover is not addressed. As a consequence, tracing methods like our ReDDM are now used in many laboratories to address intestinal homeostasis (Al Hayek et al., 2019, Arthurton et al., 2019, Martin et al., 2018, Mundorf et al., 2019, von Frieling et al., 2020).

For a key experiment (Eip75B isoforms), we now include pH3 graphs (Figure 3—figure supplement 1F) and adapted the revised manuscript accordingly.

3) Similarly, current data on changes in lipid metabolism (Figure S2C-G) should be complemented with direct assessment of gut lipid content.

We appreciate this comment and include this experiment now (Figure 2—figure supplement 1F). See reviewer 2, point 5 for details.

4) Are the metabolic and ISC phenotypes influenced by EcR signaling linked? What happens to ISC proliferation if Ec import to the Enterocytes is blocked?

We thank the reviewer for this question, which points to a logical follow up experiment to our current study. Indeed, we are currently performing these experiments in the course of another entire project. EGF and Upd ligands from EC have been implicated to induce homeostatic and stress induced signaling pathways ISC proliferation in various contexts (Jiang et al., 2011, Jiang et al., 2009, Liang et al., 2017).

So far, we can say that there is a role for systemic 20HE being translated into a local signal released from EC, as ISC proliferation is reduced upon 20HE importer RNAi (*Mex^ts^>EcI-RNAi*). This preliminary data suggests an indirect effect of 20HE acting on EC signaling to ISC, in addition to the direct role for 20HE in ISC and EB described in this current paper.

5) The use of SREBP-gal4 to manipulate gut Enterocytes (Figure 2—figure supplement 1E, F) is unconventional. Mex-gal4 or MyoIA-gal4, should be used.

We agree with the reviewer that the use of Srebp-Gal4 is unconventional and include now OilRedO staining of PMG (Figure 2—figure supplement 1I-O). In Reiff et al., 2015 we showed that *Srebp-Gal4>CD8::GFP* is the more sensitive tool to address the upregulation of lipid uptake and we were able to detect changes between VF and MF or RH5849 fed flies (Figure 2—figure supplement 1C,D,G). We agree that the measurements done in *Srebp>EcR-RNAi* might harbor some imprecision due to possible fluctuations of the driver.

The newly included OilRedO-stainings, directly addressing lipid content, show that the RH5849 agonist and the overexpression of *>EcI* significantly increase lipid content. But why is there no significant decrease when *EcR* and *EcI* are knocked down? One explanation is that there is also very low levels of OilRedO staining in PMG of controls (Figure 2—figure supplement 1I,J). Together with the Miguel-Aliaga lab, we have previously described that the uptake of lipids using OilRedO can be only visualized when egg production is blocked from instantly taking up lipids into developing eggs (Reiff et al., 2015). Therefore, we aimed to reduce egg production with *ovo^D1^*, but could not create viable stocks that harbor *ovo^D1^, Mex>* and *tub-Gal80^ts^* to perform this experiment. For future experiments, a stock directly driving GFP from the Srebp activity construct used to make Srebp-Gal4 could tackle this problem.

Taken together, we now show that in addition to the activation of Srebp, 20HE signaling activation also leads to higher lipid content directly addressed by OilRedO.

6) Images should be presented for the data in Figure 2H.

The data we presented in Figure 2H is raised with qPCR on R5 tissue to detect small differences in EcRE-lacZ mRNA levels. With images, reviewer#2 is probably referring to the Schwedes publication (Schwedes et al., 2011), in which X-Gal stainings were performed on EcRE-lacz fly stocks in different tissues. We also performed those stainings (for 1hour @37°C) for the revision and noted big differences in staining intensity among different gut regions (see Author response image 2, proventriculus). We were unable to detect differences between VF and MF (A’-B’). An intensity-based evaluation of the above images after β-Gal staining was not successful because of the signal stemming from luminal gut content (A’-C’), which is why we chose to rely on our qPCR data.

However, old flies showed strong staining in the posterior midgut suggesting an age dependency (C,C’). However, it is also possible that the known high protein stability of β-galactosidase leads to a strong signal in 20d old flies. Another explanation is that old flies have higher 20HE titers or other factors augment the response to 20HE in older flies.

7) The phenotype resulting from overexpression of E75B-A and E75B-C appears much stronger than that form EcI overexpression of the use of the EcR agonist. How do the authors explain that?

This is a very difficult question to address and we will discuss this issue based on our current data: First of all, the strong differentiation phenotype of Eip75B-A and Eip75B-C stems from the direct and forced expression in ISC/EB (*esg^ReDDM^*) or EB (*klu^ReDDM^*). Levels of *Eip75B* isoforms are probably several fold higher than under physiological conditions, but were shown to be able to rescue Eip75B LOF phenotypes in the brain (Rabinovich et al. 2016). Driven by strong *tubulin-Gal4*, we measured around 50-fold increases for all splice variants (not shown), which is much stronger than normal mating induction (Figure 3C). Overexpression with weaker drivers like *esg/klu^ReDDM^* are supposedly milder. A possible experiment to address this directly, would involve FACS sorting of *esg/klu^ReDDM^* marked cells and subsequent qPCR for Eip75B isoforms, which is not established in our lab.

Secondly, we assume 20HE ligand availability as the most probable limiting factor for the activation of EcR. In line with this, *>EcR* overexpression is unable to induce further proliferation and differentiation (Figure 1F,G). In this revised version, we now show that the overall 20HE titer in the hemolymph (Figure 2A) doubles upon mating. In line with this, we observe an about two-fold increase in progeny in *>EcI* MF (Figure 2K compared to control MF). In addition, when 20HE (and JH) titers are low in VF, EcI overexpression is not sufficient to elicit an increased number of differentiated cells. Together, we assume ligand availability as the limiting factor for EcR activation.

Feeding flies with the RH5849 non-steroidal EcR agonist affects EB to EC differentiation through Eip75B (Figure 2, Figure 6, Figure 7, Figure 7—figure supplement 1, Figure 3F) and RH5849 elicits EC production similarly to EcI MF (Figures 2Q and 7G). Chipseq, DamID and Cut-Run experiments suggest that EcR is found at between 100-1000 binding sites in the genome depending on the investigated tissue/cell line (Gauhar et al., 2009, Shlyueva et al., 2014, Uyehara and McKay, 2019). Distribution of taken up 20HE or RH5849 might additionally weaken EcR activation on a single gene level (like Eip75B) compared to direct UAS driven *Eip75B* expression. Another important trigger for activation strength of Eip75B might be an additional activation of *Eip75B* through the Notch pathway, which would be circumvented with direct >Eip75B-A/-C expression. The interplay with Notch is discussed now in the revised manuscript and under reviewer 2, point 10 and the Essential revisions.

8) Genetic interactions presented in Figure 4 should be quantified (ISC proliferation and differentiation) and controls with individual gene manipulations included.

Please see reviewer 1, point 6 and the new Figure 5 for a quantification of ISC/EB numbers and progeny numbers.

9) Can the phenotype of EcI overexpression or EcR activation be modified by knocking down E75B? Reciprocally, can E75B-A or -C overexpression rescue EcI knockdown?

We thank the reviewer for this suggestion and performed the experiments. We include the according panels and quantifications in Figure 3(F,H,K,L,M) now and discuss them accordingly in the revised manuscript.

10) The phenotype of E75B-A MARCM clones is unclear. A time course should be done. The authors claim Enteroendocrine cells are lost. However, I cannot see that in the data presented. This should be quantified. EE cell stainings are overall unclear.

We thank the reviewer for this comment and quantified EE (GFP+/Pros+) cells in *Eip75B-A^(A81)^* MARCM clones. Quantification revealed no significant difference in the number of EE compared to FRT2A controls. We noticed however that clonal EE numbers in FRT2A controls are exceptionally low (compared to FRT40A control clones, see attached quantification (A), 7d after clone induction). Additionally, we analyzed the activation of Notch in Eip75B-A clones using *GBE+Suh-dsRed* reporter flies (B-B’’, Grainy head Binding Elements and Suppressor of Hairless Notch activity reporter construct from Sarah Brays lab). We detected no significant difference in Notch signal activity that may have indicated a fate change from EC to EE (Micchelli and Perrimon, 2006, Ohlstein and Spradling, 2006, Ohlstein and Spradling, 2007). We removed the statement from the revised text.

**Author response image 3. sa2fig3:** 

11) The effect of Ec signaling activation on Notch-/- tumours does not prove a direct connection between the two phenomena. This could be the consequence of parallel action of two pathways with opposite effects. Is Ec signaling impaired in Notch tumours? This is key to connect both phenomena and substantiate a major claim of the paper.

We thank the reviewer for pointing out this important issue. Please find new data and discussion in Essential revisions 1.

Reviewer #3:This work describes the role of ecdysone signaling, especially its downstream target Eip75 in midgut homeostasis in *Drosophila.* As ecdysone spikes after mating, and the intestine was already shown to be receptive to JH-signaling from the ovary in previous work by the last author, the authors start with addressing the role of the ecdysone receptor EcR in ISC/EB proliferation and differentiation. Next, the role of the ecdysone-target Eip75B is explored. Interestingly, the analysis of different isoforms reveals different roles for different isoforms of this PPARγ orthologue, with isoforms -A and -C being involved in regulation of EB-differentiation and the -B isoform inducing ISC mitosis. The interaction with JH-signaling is investigated and lastly, the role of EcR and its downstream target Eip75B is assessed in Notch LOF-tumors. Altogether, this is an impressive volume of work that reveals a new role for the ubiquitous ecdysone hormone in both ISC proliferation and progenitor differentiation. This connects well with recent work from the Miguel-Aliaga and Perrimon labs that has explored inter-organ communication from neighbouring tissues such as fat body and ovary to the intestine. It also invites further inquiry into the interplay between mating and the Insulin-dependent growth response upon re-feeding of the midgut (O'Brien et al., 2011). I would recommend this work for publication after the issues described below would be addressed. Especially in the light of the current Covid19-crisis, I do not think extra experimental work should be necessary.Major comments:1) Central to their thesis is the fact that ecdysone produced in the ovary would signal to ISCs/EBs to exert its effect. However, the authors have not done any genetic experiments to show that flies without ecdysone production in the ovary would not have the mating-induced response ISC-EB increase. Hence, a role of local ecdysone production by either the fat body or gut cannot be ruled out. Related to this, the authors decribe determination of 20HE-titers on wild-type and ovoD1 heterozygotes in their Materials and methos section, but no data can be found in the manuscript. Can the authors comment on that and include the 20HE-titer data?

We thank reviewer for pointing to this central issue as well and added this data now. Please find new data and discussion in Essential revisions 1.

2) This relates more to the parallels drawn between the work described here and the parallels with mammalian steroid hormone signaling. Based on their results, I do not think the statement that "our findings suggest a tumor-suppressive role for steroidal signalling " is warranted. Loss of EcR suppresses ISC/EB proliferation upon mating and the EcR agonist RH5849 dramatically induces proliferation, which would be more in line with an oncogenic effect of EcR-signaling in the fly. I understand that EcR induces Eip75, a PPARγ orthologue and they provide evidence that this induces differentiation and therefore impairs Notch LOF-driven tumorigenesis, in line with mammalian data. I would suggest altering some of these statements and moving them out of the Abstract and Introductionto the Discussion section, where there is more room to elaborate on these contradictions.

We followed the reviewer(s) suggestion here and moved and softened statements in Abstract, Introduction and the final paragraph of the Results section. Furthermore, we added experiments with an PPARy agonist impairing Notch driven tumorigenesis and extended and focusing the discussion accordingly (new Figure 4).

Concerning a role for steroidal signaling and tumor suppressive or oncogenic roles: the effects of ER signaling in humans are context dependent. The major functions of steroid hormone receptor signaling in normal adult tissues appear to involve differentiation rather than proliferation (Cheng and Balk, 2003). This role is drastically changed in the pathology of cancer and there is a plethora of studies showing the benefits of blocking ER signaling in breast and prostate cancer (Cheng and Balk, 2003), whereas activation seems to be beneficial for female colorectal cancer patients (Chen et al., 1998, Hendifar et al., 2009, Lin et al., 2012).